# Quantum transport protected by acceleration from nonadiabaticity and dissipation

Arnab Chakrabarti [1,2,8] ✉, Biswarup Ash[3,4,8], Igor Mazets [5,6], Xi Chen [7] & Gershon Kurizki [1]

We put forth a hitherto unexplored control strategy that enables high-fidelity fast transport of an unstable quantum wavepacket even in the presence of bath-induced dissipation. The wavepacket, which is confined within any shallow (anharmonic) potential trap is steered in acceleration, so as to maximize the transfer fidelity. This strategy can generally optimize any non-Markovian bath-dressed continuous-variable system dynamics. It can simultaneously cope with wavepacket leakage via non-adiabatic transitions and bath-induced dissipation in an optimal fashion. It can outperform methods based on counterdiabatic fields (shortcuts to adiabaticity) particularly for fast non-adiabatic transport. Transport fidelity is maximized even for trajectories exceeding the speed of bath-excitation propagation, e.g., for supersonic transfer through phonon baths. This general approach is illustrated for optimized transfer of impurities in Bose-Einstein condensates. It is applicable to both dissipative and non-dissipative transfer of trapped atoms and ions and molecular reaction products.

The ability to minimize the relaxation and decoherence of open quantum systems, so as to protect the fidelity of their coherent evolution, is a major challenge of quantum science and technology, at both fundamental and applied levels[1–3]. The main thrust has been on decoherence control of discrete variables in qubit systems because of their central role in quantum information processing[1]. Such control calls for intervention in the system evolution on non-Markovian time scales[3–6].

Here we address the much less explored task of suppressing the leakage of unstable quantum wavepackets from finite-depth (hence anharmonic) trapping potentials to the continuum, a process akin to tunneling in nuclear alpha decay[7] and atomic traps[8,9] or in superconducting devices[10]. The spread of stable quantum wavepackets can be suppressed by a resonant drive[11]. However, this method does not apply to the scenarios discussed here.

Because of the unrestricted number of degrees of freedom involved, the task of maintaining high fidelity of the initial wavepacket becomes much more challenging when the wavepacket is moving through a dissipative environment. Pertinent scenarios involve a trapped multiatom impurity moving through a condensate as a Bose polaron[12–14], or as a part of a quantum refrigeration cycle[15]; transport of ions in a trap[16–19], trapped-atoms transported in vacuum (e.g., using tweezers)[20,21] where dissipation vanishes but non-adiabatic leakage from the trap hampers fidelity. Analogous scenarios arise in molecular dissociation or collisions[22,23] where the wavepacket is moving along a potential surface while being dissipated by other (rovibronic or electronic) degrees of freedom[24,25].

The fundamental dilemma in such scenarios is that the faster is the wavepacket transferred the less it is affected by the environment

[1]AMOS and Department of Chemical and Biological Physics, Weizmann Institute of Science, Rehovot, Israel. [2]Department of Physics, Rajiv Gandhi University, Rono Hills, Doimukh, Arunachal Pradesh, India. [3]Department of Physics of Complex Systems, Weizmann Institute of Science, Rehovot, Israel. [4]Department of Physics, University of Michigan, Ann Arbor, MI, USA. [5]Vienna Center for Quantum Science and Technology (VCQ), Atominstitut, TU Wien, Vienna, Austria. [6]Wolfgang Pauli Institut c/o Fakultät für Mathematik, Universität Wien, Vienna, Austria. [7]Instituto de Ciencia de Materiales de Madrid (CSIC), Cantoblanco, Madrid, Spain. [8]These authors contributed equally: Arnab Chakrabarti, Biswarup Ash. ✉e-mail: arnab.chakrabarti@rgu.ac.in

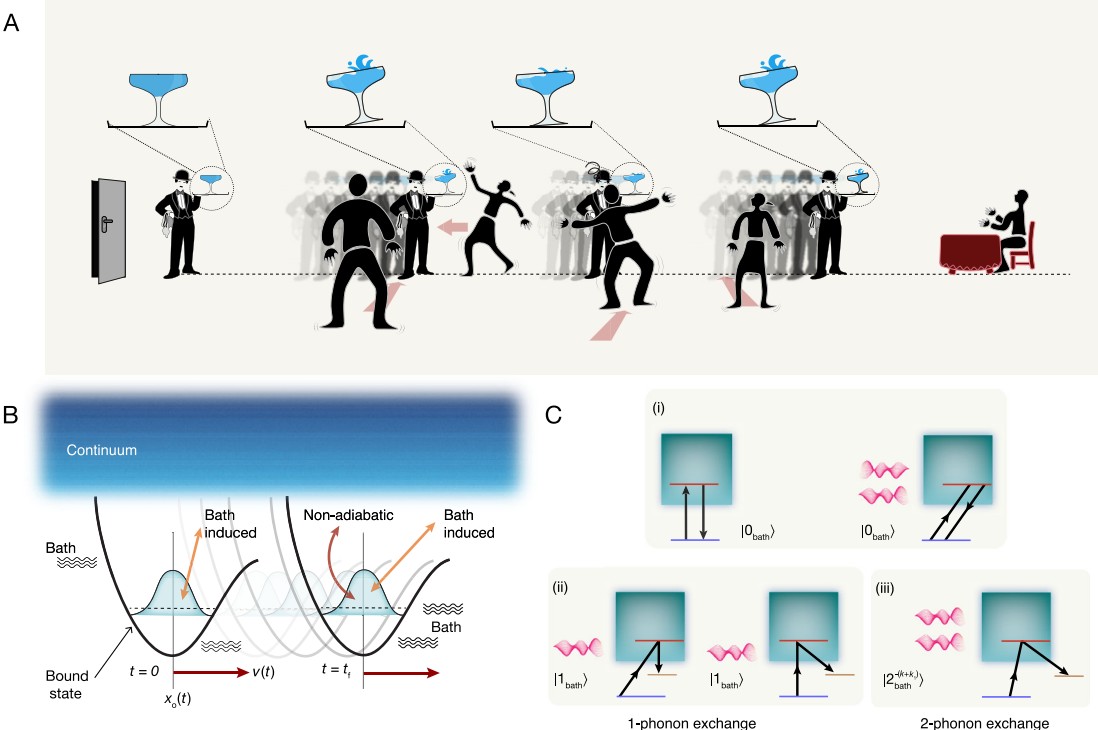

**Fig. 1 | Illustration of the quantum transport problem. A** Classical analog of our problem: A waiter carrying a shallow glass filled up to the brim with water, in a crowded hall. Random kicks from the crowd makes water spill out of the glass even when the waiter is static. Moving with a tilted tray without spilling water from such an unstable system is almost impossible. Neither is evasive maneuvering by the waiter which requires frequent feedback. Instead, the waiter can speed up as much as possible to avoid kicks, with occasional slowdown to keep the energy constraint. **B** Schematic view of nonadiabatic transport of a quantum wavepacket through a dissipative medium, in a shallow moving trap or motion along a dissipative potential surface. **C** Self-energy diagrams contributing to leakage and dissipation: (i) No quanta exchange (purely non-adiabatic−left panel) and virtual quanta exchange (purely bath-mediated−right panel) with the bath. (ii) Real quanta exchange via partly non-adiabatic and partly bath-induced processes (iii) Real quanta exchange with the bath via excitation non-conserving processes. In panels (i), (ii) and (iii) the horizontally shifted levels indicate advanced (scattered) wave-packet states, the vertical arrows indicate non-adiabatic transitions, while the oblique arrows indicate bath-induced transitions.

(bath), but, on the other hand, its nonadiabatic evolution increases its leakage out of the finite-depth trap. To prevent such leakage, the standard recipe that comes to mind is the use of shortcuts to adiabaticity (STA), either by applying counterdiabatic fields (CDF) or by inverse engineering of the system Hamiltonian based on dynamical invariants[26–37]. For continuous variable systems such as trapped wavepackets, only dynamical invariants quadratic in momentum are useful[30]. However, such invariants only exist for the special Lewis-Leach class of potentials[28,30,38,39]. Consequently, for wavepackets in arbitrary trapping potentials, invariant based STA methods are hardly applicable.

In general, STA is mainly geared to closed, stable quantum systems[35] since, being a hamiltonian method, it is apriori unclear, to what extent can STA suppress irreversible bath effects[35,40,41]? One should be mindful that the success of any Hamiltonian protocol for a lossy quantum system depends on the typical ratio of the level-width to level-spacing. Yet, there have been numerous extensions of STA techniques to open quantum systems, mostly of either discrete variables or harmonic potentials (invariants quadratic in momentum)[42–62]. Application of the invariant based STA in open quantum systems, as in[42–50] has limited applicability for arbitrary (non-harmonic) trapping potentials. Extensions of the CDF method to transitionless driving of open quantum systems, typically composed of discrete variables, may involve non-Hamiltonian or non-Hermitian control[51,59], as opposed to the unitary control for closed-system STA.

On top of the difficulties in using STA for general open quantum systems, none of the above methods can simultaneously control the irreversible leakage of an unstable wavepacket due to combined non-

adiabatic and bath-induced transitions in a realistic model, where the time-dependence of the system Hamiltonian induces changes in the bath-induced leakage. This is the challenging problem we address in this work.

To illustrate the difficulty, it is instructive to consider the classical analog of the problem: a waiter carrying a shallow glass of water filled to the brim (Fig. 1A) in a crowded hall. Random kicks can cause water to be spilled out of the glass even when the waiter stands still, let alone moves. The compensating counter-diabatic field (CDF) is tantamount to tilting the tray[27]. Yet such tilting cannot prevent the waiter from spilling out the water, the glass being a randomly-perturbed unstable system.

Here we advance an altogether different strategy, dubbed acceleration-controlled quantum dissipative transport (AC-QUDIT), which we rigorously show to be more effective than STA for some randomly-kicked unstable quantum systems, carried over a broad range of transport speeds: to minimize the spilling, the waiter should move around at a variable pace on non-Markovian time scales, i.e., change velocity faster than the correlation time of the random kicks. We show how such pace control should be executed, if instead of water the shallow vessel would contain a quantum liquid or wavepacket.

To this end we introduce motion control of arbitrary dissipating wavepackets by a general non-Markovian description of coupled system-bath quantum dynamics as in the Wigner-Weisskopf description[63,64], previously used by our group in discrete-variable control[8,65]. Euler-Lagrange (EL) optimization then yields a non-Markovian integro-differential equation of motion of the wavepacket, that accounts for both non-

adiabatic and bath effects. For a general Frölich-type coupling[12,13,66], this results in a nonlinear integro-differential equation which is analytically solvable in a speed regime that, remarkably, allows for significantly non-adiabatic transfer and can even be faster than the bath-excitation propagation, e.g., take place at a supersonic speed in phonon baths. The more non-adiabatic the motion, the more advantageous is AC-QUDIT compared to STA. The advocated approach is broadly applicable to atomic and molecular quantum wavepacket transport in the scenarios mentioned above[12–15,19–25].

## Results

### General scenario

We consider a wavepacket describing a quantum single- or multi-partite system of mass $m$, in a trapping potential $V[x - x_\circ(t)]$, whose center (minimum) position $x_\circ(t)$ is driven by an external force. It is immeresed in a bath of interacting or free bosons of mass $m_B$ (Fig. 1B). The system + bath compound is described by the time-dependent Hamiltonian

$$H(t) = H_S(t) + H_B + H_{SB},$$
$$H_S(t) = \frac{p^2}{2m} + V[x - x_\circ(t)] ; H_B = \sum_{k \neq 0} \Omega_k b_k^\dagger b_k. \qquad (1)$$

Here $b_k^\dagger$ and $b_k$ are the creation and anihilation operators for a bath excitation, e.g., a phonon with frequency $\Omega_k$ and momentum $k$, while $x$ and $p$ denote the position and conjugate momentum of the system.

Rather generally, the quantized system-bath interaction Hamiltonian, without resorting to the Lamb-Dicke limit[67], nor to the rotating wave approximation[64], is taken to be of the Frölich-type[12,13,66]:

$$H_{SB} = \sum_{k \neq 0} [g_k \, b_{-k} \, e^{-ikx} + h.c.], \qquad (2)$$

$g_k$ being the $k$-mode coupling-constant (see "Methods", SI). Here, the system operator $e^{-ikx}$ imparts an overall momentum of $-k$ to the instantaneous wave-packet while the operator $b_{-k}$ simultaneously annihilates a phonon having the same momentum $-k$, thereby ensuring conservation of momentum (see SI IV, "Methods").

For simplicity, the trap $V[x - x_\circ(t)]$ is assumed to be shallow, such that it supports a single bound-state along with a continuum of unbound (scattering) states, although this assumption is non-essential. Then, under non-adiabatic and bath-induced transitions, the system dynamics is similar to the Friedrichs model for resonance phenomena[68].

### Transported wavepacket dynamics

The system-bath dynamics can be described by an expansion of the combined system + bath state at time $t$ as:

$$|\psi(t)\rangle = \sum_\alpha A_\alpha(t) \, e^{-i\omega_\alpha t} \, |\alpha(t)\rangle + \sum_\beta A_\beta(t) \, e^{-i\omega_\beta t} \, |\beta(t)\rangle. \qquad (3)$$

Here $\{|\alpha(t)\rangle\}$ and $\{|\beta(t)\rangle\}$ denote the instatenous Fock product basis-states of $H_S(t) + H_B$[12], corresponding to bound and unbound instantaneous energy eigenstates of the system, respectively. For any $l \in \{\{\alpha\}, \{\beta\}\}$, $\omega_l$ denotes the eigen-energy ($\hbar = 1$) and $A_l(t)$ the instantaneous probability amplitude of the bound or unbound state $|l(t)\rangle$. The instantaneous energy eigenstates are time-dependent, but the corresponding instantaneous eigen-energies are independent of time for a given potential $V(x)$. In this instantaneous eigenbasis, the bound-to-continuum transition amplitudes of the system are dependent on the center-of-mass position $x_\circ(t)$ and velocity (speed) $\dot{x}_\circ(t)$ (see SI V). We can then design an optimal control of the trajectory $x_\circ(t)$ so as to maximize the bound-state transfer fidelity.

We label the instantaneous bound and scattering states of the moving trap by $|n(t)\rangle$ and $|\epsilon(t)\rangle$, with $\epsilon > 0$, which are functions of $x - x_\circ(t)$. At the start of the quench, the system is assumed to be in the

bound state of potential with no excitation in the bath, denoted by $|\nu(0)\rangle = |n(0)\rangle \otimes |0_{\text{bath}}\rangle$, where $|0_{\text{bath}}\rangle$ indicates the vacuum bath state. The Schrödinger equation $i \frac{\partial}{\partial t} |\psi(t)\rangle = H(t)|\psi(t)\rangle$ then yields the set of equations

$$\dot{A}_l = -\sum_j A_j R_{lj}(t) - i \sum_j A_j S_{lj}(t), \qquad \forall l, j \in \{\{\alpha\}, \{\beta\}\}. \qquad (4)$$

In the first term on the r.h.s. of Eq. (4) $R_{lj}(t) = \langle l(t)| \frac{\partial}{\partial t} |j(t)\rangle$ represent non-adiabatic transition rates due to the motion while in the second term $S_{lj}(t) = \langle l(t)|H_{SB}|j(t)\rangle$ denote the bath-mediated transitions induced by the coupling $H_{SB}$.

At $t > 0$, the system-phonon scattering entangles advanced / retarded and instantaneous bound and excited states of the system wavepacket and the many-body bath states, which are required for momentum conservation (see SI IV). The survival probability (fidelity) of the instantaneous bound state of the trapped wavepacket is the same as the Loschmidt-echo probability[12,69] of finding the system in the instantaneous bound state with no bath excitations, $|\nu(t)\rangle = |n(t)\rangle \otimes |0_{\text{bath}}\rangle$, at time $t$ (see "Methods", SI IV, VI). To evaluate this probability, we integrate out all probability amplitudes in the bound and unbound sectors except that of the initial state.

In our non-perturbative approximation we retain only terms of leading order in $\dot{x}_\circ$ and $H_{SB}$, in the rate of change of the survival probability amplitude of the initial wavepacket[70,71]. This standard approximation in decay theory is valid whenever the coupling strengths (system-bath + non-adiabatic) are weaker than the inverse time-scales of the corresponding reservoir (bath or continuum) dynamics[4,8,70–75] (see SI V). Then the self-energy diagrams for the rate of change of the Loschmidt echo amplitude consist of only diagrams shown in Fig. 1C (i). These include simultaneous system-and-bath (de) excitations which are negligible at long time-scales but are important at short times, when the system energy-uncertainities are substantial. The resulting expression for the survival probability at $t = t_f$, obtained from the Wigner-Weisskopf method, is a resummation of all second-order processes in Fig. 1C (i)[12,76], akin to a Dyson series and hence represents a non-linear response (see SI V):

$$\mathcal{P}(t_f) = \exp(-J[x_\circ, \dot{x}_\circ]), \qquad (5)$$

$J[x_\circ, \dot{x}_\circ] =$

$$\text{Re} \int d\epsilon \left[ \int_0^{t_f} dt_1 \int_0^{t_f} dt_2 \left\{ \underbrace{\gamma_{n\epsilon}(t_1) \, \gamma_{n\epsilon}^*(t_2)}_{\text{non-adiabatic}} + \frac{L}{2\pi} \int dk \, \underbrace{\Delta_{n\epsilon}^k(t_1) \, \Delta_{n\epsilon}^{k*}(t_2)}_{\text{bath-induced}} \right\} \right]. \qquad (6)$$

Here the integrals over $\epsilon$ and $k$ indicate the cumulative effects of all $n \rightarrow \epsilon$ (bound-to-continuum) transitions and all $k$-modes of the bath respectively and $L$ is the confining length of the medium (bath). The first term on the r.h.s. of Eq. (6) describes non-adiabatic transitions in Fig. 1 C (i-left) in terms of the leakage rates $\gamma_{n\epsilon}(t) = e^{-i\omega_{\epsilon n} t} \langle n(t)|\dot{H}_S(t)|\epsilon(t)\rangle / \omega_{\epsilon n} = \dot{x}_\circ(t) \left(\frac{\mu_{n\epsilon}}{\omega_{\epsilon n}}\right) e^{-i\omega_{\epsilon n} t}$ where $\left(\frac{\mu_{n\epsilon}}{\omega_{\epsilon n}}\right)$ denotes the nonadiabatic coupling strength with frequencies $\omega_{\epsilon n} = \omega_\epsilon - \omega_n$ and $\mu_{n\epsilon} = -\langle n(t)|\frac{\partial V(q)}{\partial q}|\epsilon(t)\rangle$; $q = x - x_\circ(t)$. The second term describes the bath-mediated processes in Fig. 1C (i-right) in terms of the transition rates that are proportional to the system-bath coupling strengths $g_k$ for bath mode $k$: $\Delta_{n\epsilon}^k(t) = e^{-ikx_\circ(t)} g_k \langle n(t)| e^{-ikq} |\epsilon(t)\rangle e^{-i(\Omega_k + \omega_{\epsilon n}) t}$ integrated over bath-mode wave-vectors $k$. The survival probability for dissipationless transport corresponds to the vanishing of the bath-induced terms in Eq. (6). In deriving the above expression, we have used the fact that the volume of $k$-space per allowed value of $k$ is $\Delta k = 2\pi/L$.

Upon resorting to the finite-time Fourier transform $f_{t_f}(\omega) = \int_0^{t_f} d\tau \, e^{-i\omega\tau} f(\tau)$, we can rewrite the first (non-adiabatic) term

on the r.h.s. of (6) as : $\int d\epsilon |\frac{\mu_{n\epsilon}}{\omega_{en}}|^2 |(\dot{x}_o)_{t_f}(\omega_{en})|^2 \geq 0$ while the second (bath-induced) term becomes $\frac{1}{2\pi} \int d\epsilon \, dk \, |(\Delta_{n\epsilon}^k)_{t_f}(\omega_{en})|^2 \geq 0$. The positivity of both terms confirms the decay of the survival probability in time. The non-adiabatic contribution to $J[x_o, \dot{x}_o]$ is identified as the power-spectral density (PSD) of the trap speed, integrated over the entire range of the continuum index $\epsilon$ of the system.

## Control of nonadiabatic and dissipation losses by optimized acceleration

The maximum survival probability at $t = t_f$ corresponds to the minimized value of $J[x_o, \dot{x}_o]$, subject to a physical constraint, chosen here to be related to the kinetic energy $E_K$ supplied for the transport[5]. For a given $E_K$, the Lagrange multiplier $\lambda$ determines the total cost functional for Euler-Lagrange (EL) optimization as

$$J_{\text{tot}}[x_o, \dot{x}_o] = J[x_o, \dot{x}_o] + \lambda J_1[\dot{x}_o];$$
$$J_1[\dot{x}_o] = \int_0^{t_f} d\tau \, [\dot{x}_o(\tau)]^2 - E_K. \tag{7}$$

The stationarity condition $\delta J_{\text{tot}} = 0$ results in a non-linear and non-local EL equation, obtained from Eqs. ((6)–(7)) (see SI VII). This non-linear integro-differential equation ("Methods") cannot be solved exactly. Its brute-force numerical solution can be impractical in many cases. Approximate analytical solutions are only possible in the speed range, $|\dot{x}_o(t)| < v_s = |(\omega_{en} + \Omega_k)/k|$ for a given transition frequency $\omega_{en}$ and Fourier harmonic $\Omega_k$ of the $k$-mode bath response (see "Methods", SI XI, Supplementary Fig. S-3). We then obtain to lowest order in this high-speed factor, the following linearized integro-differential EL equation which is our central result ("Methods"):

$$\lambda \ddot{v}(t) = -\eta(t) - \zeta(t) v(t) + \int_0^{t_f} d\tau \, \phi(t - \tau) v(\tau). \tag{8}$$

Here $v(t) = \dot{x}_o(t)$, $\eta(t) = \int_0^{t_f} ds \frac{L}{2\pi} \int d\epsilon \, dk \, (\omega_{en} + \Omega_k) k \, |\tilde{g}_{n\epsilon}^{(k)}|^2 \cos[(\omega_{en} + \Omega_k)(t - s)]$, $\zeta(t) = \int_0^{t_f} ds \frac{L}{2\pi} \int d\epsilon \, dk \, k^2 \, |\tilde{g}_{n\epsilon}^{(k)}|^2 \cos[(\omega_{en} + \Omega_k)(t - s)]$, $\tilde{g}_{n\epsilon}^{(k)} = g_k \langle n(t)|e^{-ikq}|\epsilon(t)\rangle$ and $\phi(t) = \int d\epsilon |\mu_{n\epsilon}|^2 \cos[\omega_{en} t]$. We have arrived at this linear integro-differential equation (8) from the fully non-linear EL equation obtained from minimization of Eq. (7), by a method adopted in frequency discriminator circuits for FM demodulation[77].

Equation (8) determines the optimal acceleration control of a dissipated, nonadiabatic moving trapped wavepacket. It involves a major generalization of the universal Kofman-Kurizki (KK) formula of non-Markovian dynamical control for discrete variables[8,22]. In the K-K formula the control field only acts as an amplitude modulation filter of the bath response spectrum whereas, here $\ddot{v} = \dddot{x}_o$ converts frequency variations of $k\dot{x}_o(t)$ to amplitude variations of $kv(t)$ in the response function which is proportional to $\zeta(t)$ while $\phi(t)$ undergoes the usual amplitude filtering via $v(t)$.

The salient merit of our AC-QUDIT method is the ability to optimally protect the fidelity against both non-adiabatic leakage and dissipation simultaneously, by trajectory control. We find this optimal trajectory by solving Eq. (8), under boundary conditions $x_o(0) = 0$, $v(0) = 0$, while allowing for finite final trap-speed $v(t_f)$. This choice of boundary conditions uniquely determines the optimal trajectory $x_o(t)$, thereby fixing the final trap position $x_o(t_f)$ (see SI X, XII, Supplementary Fig. S-4.). Importantly, even the linearized EL equation (8) depends on $kv(t)$ as we venture beyond the Lamb-Dicke regime $|kx_o| \ll 1$. In fact $kx_o$-nonlinearity is essential for effective control of bath-induced dissipation (Methods).

In the case of dissipationless transport of the wavepacket through vacuum or near-vacuum, as explored in[20,21] (i.e., zero system-bath coupling), the survival probability in Eq. (5) becomes a functional of the trap velocity only, since $\Delta_{n\epsilon}^k(t)$ vanishes in (6). In this case $v(t)$ serves as our control parameter instead of $x_o(t)$ and the corresponding optimal control problem is designed by imposing constraints on the control bandwidth[5]: $J_1[v] = \int_0^{t_f} dt_1 [\dot{v}(t_1)]^2 - E$ and the total distance traveled: $J_2[v] = \int_0^{t_f} dt_1 v(t_1) - S$ [fixing the final trap-position $x_o(t_f)$] for some constants $E$ and $S$.

Introducing Lagrange multipliers $\lambda, \lambda_1$, the stationarity of the total cost functional $J_{\text{tot}}[v] = J[v] + \lambda J_1[v] + \lambda_1 J_2[v]$ then yields the EL equation:

$$\lambda \ddot{v}(t) = \frac{1}{2}\lambda_1 + \int_0^{t_f} ds \, \phi_1(t - s) v(s) \tag{9}$$

where $\phi_1(t) = \int d\epsilon \, |\frac{\mu_{n\epsilon}}{\omega_{en}}|^2 \cos(\omega_{en} t)$. For $\lambda_1 \neq 0$, Eq. (9) is a linear Fredholm integro-differential equation of the second kind which can be solved using methods adopted for solving Eq. (8) (SI X).

## Comparison with STA

A compensating counter-diabatic field (CDF), $\dot{x}_o(t) p$, is added in the STA approach, to the system Hamiltonian (1) in order to achieve transitionless transport of the trapped particle[27,78]. Such a CDF that breaks time-reversal symmetry, is hard to implement experimentally. Instead, a gauge transformation of the form $p \rightarrow p' = p + \partial_x f$, $H_{CDF} \rightarrow H'_{CDF} = H_{CDF} + \partial_t f$ with $f(x, t) = -m\dot{x}_o x$, is usually invoked as a more practical compensating term in the form of an effective gravitational field: $-m\ddot{x}_o x$[27,78]. Here $H_{CDF}$ indicates the system Hamiltonian with a CDF term.

Such local gauge transformations require a corresponding transformation of the wave functions: $\psi(x, t) \rightarrow \psi(x, t)' = e^{-if(x, t)} \psi(x, t)$. If $\psi(x, t)$ denotes an eigenstate of the untransformed system Hamiltonian $H_S$, the corresponding transformed state $\psi(x, t)'$ is not an eigenfunction of $H'_S$ unless $\partial_t f = 0$. Namely, only in the case of uniform trap speed, does the gauge transformation preserve the eigenstates. Thus, for an accelerating trap, a particle initialized in a bound state of the initial trap position continues to experience non-adiabatic transitions even in presence of the compensating gravitational field.

If the trapping potential is shallow and admits a continuum spectrum, such non-adiabatic transitions lead to irreversible loss of fidelity even in the absence of an external heat bath[8–10,79]. This irreversible loss of fidelity cannot be avoided by merely making the velocity and acceleration vanish at the boundary points, as is usually done to achieve transition-less transport via STA[27,78] (see "Methods").

The introduction of a compensating counter-diabatic field (CDF) changes the non-adiabatic transition rates $\gamma_{n\epsilon}(t)$ in Eq. (6) as follows :

$$\Gamma_{n\epsilon}(t) = \gamma_{n\epsilon}(t) - im\dot{v}(t)D_{n\epsilon}e^{-i\omega_{en}t}. \tag{10}$$

where $D_{n\epsilon} = \langle n(t)| q |\epsilon(t)\rangle$, is the time-independent dipole transition matrix element. By contrast, $\Delta_{n\epsilon}^k(t)$ remains invariant since CDF does not involve bath-induced transitions. The first term on the r.h.s. of (6) is then transformed to Re $\int d\epsilon \int_0^{t_f} dt_1 \int_0^{t_f} dt_2 \, \Gamma_{n\epsilon}(t_1)\Gamma_{n\epsilon}^*(t_2) = \int d\epsilon | \int_0^{t_f} dt_1 \Gamma_{n\epsilon}(t_1)|^2 > 0$, which can be rewritten as a sum of terms proportional to the trap-speed PSD, corresponding trap-acceleration PSD and the cross-spectral density (CSD) between the trap-speed and acceleration, albeit the overall integral over $\epsilon$ (see "Methods" for detailed expressions).

In general the CDF method would yield lower values of $\mathcal{P}(t_f)$ than that obtained by our approach whenever

$$\int_0^{t_f} dt_1 \int_0^{t_f} dt_2 \, \Gamma_{n\epsilon}(t_1)\Gamma_{n\epsilon}^*(t_2) > \int_0^{t_f} dt_1 \int_0^{t_f} dt_2 \, \gamma_{n\epsilon}(t_1)\gamma_{n\epsilon}^*(t_2), \tag{11}$$

which holds provided the sum of the speed-acceleration CSD and the acceleration PSD is positive (see "Methods").

## Numerical results

While the AC-QUDIT method described by the linearized Eq. (8) is general, with broad applicability in AMO physics as it can be used for any trapping potential and bosonic bath, we have numerically illustrated the method for an impurity in a moving Morse-potential trap immersed in a BEC (see "Methods" and SI I -- V, VIII, IX). Expressing all length scales in units of the phonon coherence length $\xi$, mass in units of the phonon mass $m_B$ and time in units of the characteristic time

$t_B = \xi/\sqrt{2}c$ (see "Methods", Table 1) where $c$ is the speed of sound in BEC[74], we study the dependence of the survival probability (fidelity) $\mathcal{P}(t_f)$ on the final transport time $t_f$ for different values of the impurity mass $m$ and final trap speeds $v(t_f)$. Survival probabilities at transport durations shorter than the characteristic time of non-adiabatic leakage $t_c$ ("Methods") as well as $t_B$ are relevant for quantum information processing.

Analytical results of Eq. (8), confirmed by numerical solutions of the fully non-linear Eq. (12) (see "Methods", SI XI, XIII, Supplementary Fig. S-1) show that AC-QUDIT invariably ensures higher fidelity than constant-velocity transport, all parameters being equal. In the absence of dissipation (e.g., for atoms trapped in a tweezer moving in vacuum) the advantage of AC-QUDIT becomes salient for appreciable non-adiabatic leakage rates (Fig. 2A) (Methods). Under bath-induced dissipation, AC-QUDIT is more advantageous compared to the constant velocity transport, for faster (less adiabatic) transfer (Fig. 2 B). The final speed $v(t_f)$ and the constant speed $\tilde{v}$ are here chosen to be non-adiabatic when they are higher than $c$, the speed of sound in the BEC (see "Methods").

Defining the $k$-independent coupling strength $\tilde{g}$ as $g_k = \tilde{g} U(k)$, $U(k)$ being a suitable function of $k$ (Methods), we find that an increase in $\tilde{g}$ i.e., an increase in the bath-induced rates $\Delta_{nc}^k$ results in a corresponding decrease in the survival probability.

## Table 1 | Scaled and unscaled parameters

| Parameter | Unscaled | Scaled |
|---|---|---|
| Trap-depth | $D$ | $D' = Dt_B$ |
| Impurity mass | $m$ | $m' = \frac{m}{m_B}$ |
| Trap-width parameter | $a$ | $a' = a\xi$ |
| Continuum correlation time | $t_c$ | $t_c' = \frac{t_c}{t_B}$ |
| Phonon wave vector | $k$ | $k' = k\xi$ |
| Phonon coupling strength | $g_k$ | $g_{k'}' = g_{k\xi} t_B$ |

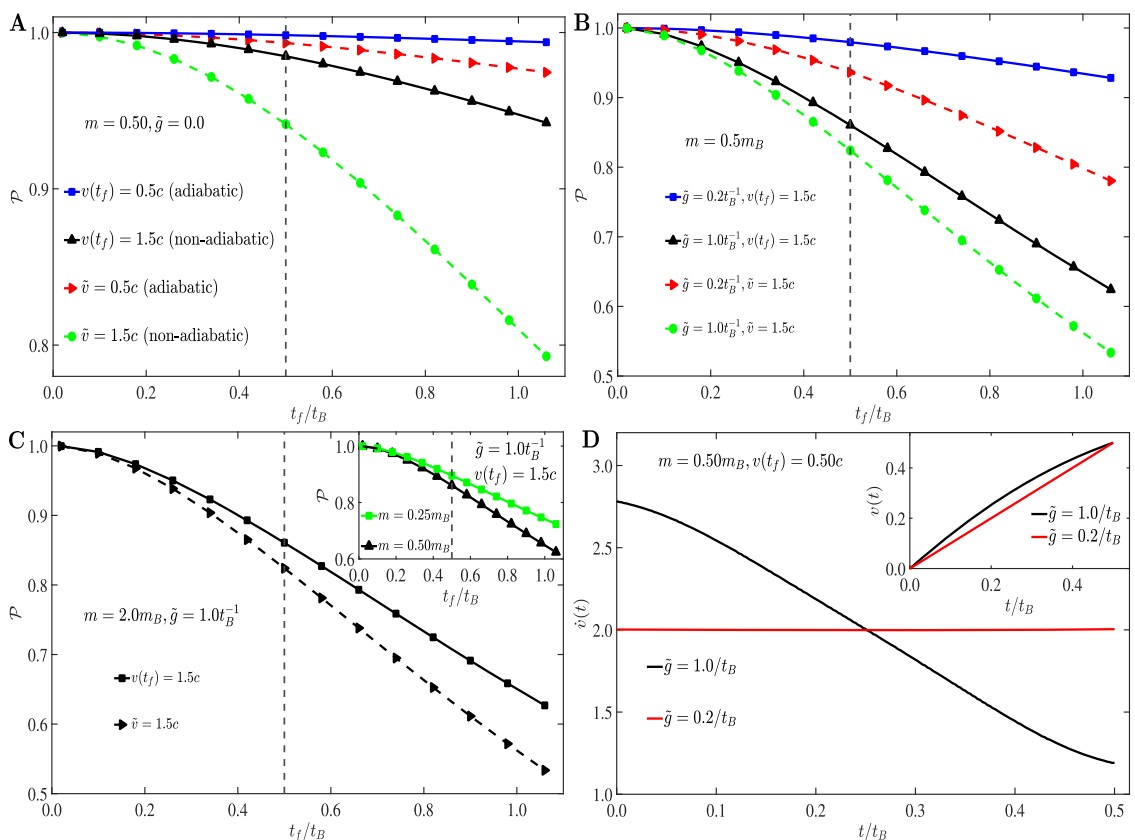

**Fig. 2 | Comparison of the AC-QUDIT method with constant speed transport.**
**A** Survival probabilities $\mathcal{P}(t_f/t_B)$ for dissipation-less transport (through vacuum) at non-adiabatic final speed $v(t_f / t_B) = 1.5c$ (black) as well as adiabatic final speed $v(t_f / t_B) = 0.5c$ (blue) for $m = 0.5\,m_B$, obtained by our AC-QUDIT method compared to corresponding constant speed motion with speeds $\tilde{v} = 1.5c$ (green, non-adiabatic) and $\tilde{v} = 0.5c$ (red, adiabatic). In BEC, $c$ is the speed of sound. In vacuum it can be replaced by $1/2at_c$ where $a$ is the Morse trap-width parameter and $t_c$ the coherence time associated with nonadiabatic leakage from the trap (Methods), $m$ can be measured in any arbitrary unit while for simplicity we have replaced $t_B$ by $2t_c$. Throughout, we have used the Lagrange multipliers $\lambda = 1$, $\lambda_1 = 1$. **B** Survival probabilities for transport through BEC (dissipative: $\tilde{g} = 1.0t_B^{-1}$ and $\tilde{g} = 0.2t_B^{-1}$) at

non-adiabatic and adiabatic final speeds $v(t_f/t_B) = 1.5c$ and $0.5c$ respectively for $m = 0.5m_B$ obtained by our AC-QUDIT method (solid lines) compared to constant speed motion with $\tilde{v} = 1.5c$ and $0.5c$ (dashed lines). **C** Survival probabilities obtained by AC-QUDIT method for $m = 2.0m_B$ (heavier impurity) and $v(t_f/t_B) = 1.5c$ compared with constant speed transfer at $\tilde{v} = 1.5c$ (see "Methods"). Inset shows the survival probability as a function of the transport duration for $m = 0.25 m_B$ and $m = 0.5 m_B$ (lighter impurities), all other parameters remaining same in this case (see "Methods"). In **A**–**C** the vertical dashed line indicates the $t_f/t_B = t_c/t_B$ point on the horizontal axis. **D** Optimal trap acceleration obtained from our method, as a function of time, for $\tilde{g} = 0.2\, t_B^{-1}$ (red line) and $\tilde{g} = 1.0\, t_B^{-1}$ (black line) for $v(t_f / t_B) = 1.5\, c$, $m = 0.5\, m_B$. The inset shows the corresponding optimal trap-speeds.

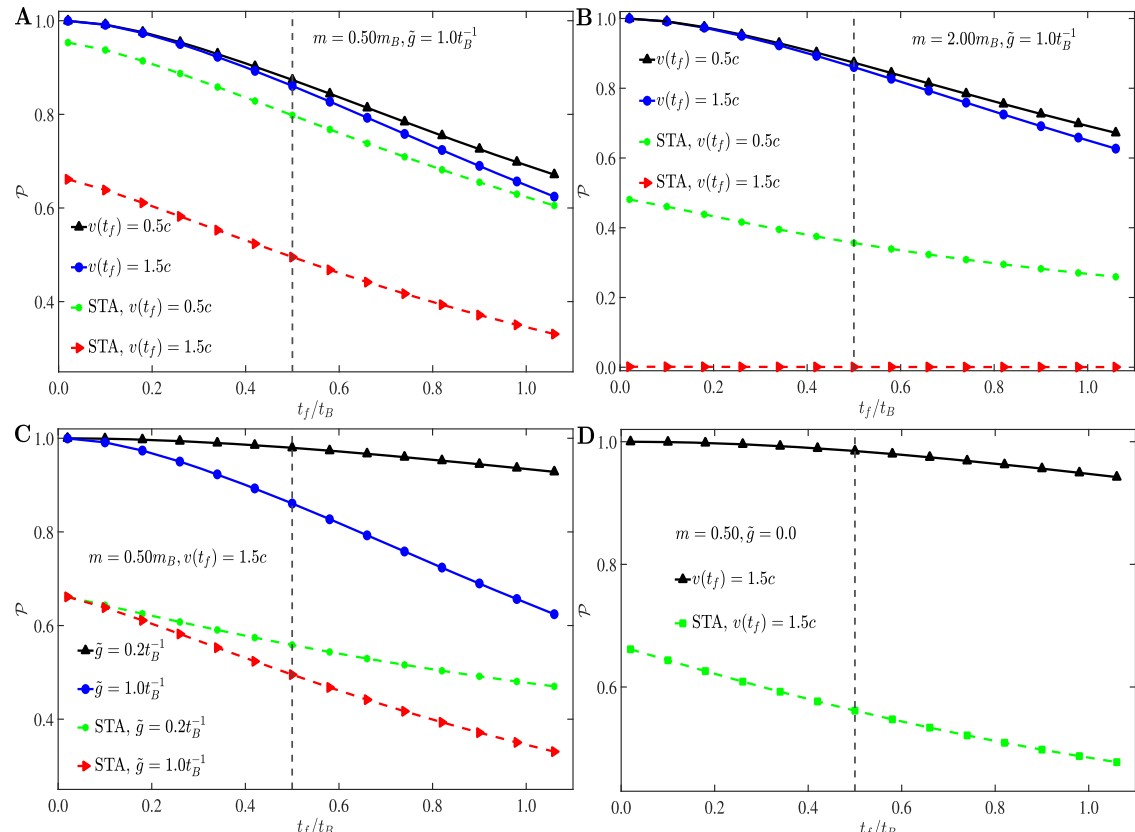

**Fig. 3 | Comparison of the AC-QUDIT method with Shortcut to Adiabaticity (STA) using a CDF. A** Survival probabilities as a function of $t_f/t_B$ obtained by CDF (STA) (dashed lines) and AC-QUDIT (solid lines) for $m = 0.5\,m_B$, $v(t_f/t_B) = 0.50\,c$ and $v(t_f/t_B) = 1.5\,c$. **B** Same as (**A**) with $m = 2\,m_B$ and final speeds $v(t_f/t_B) = 0.5c$ and $v(t_f/t_B) = 1.5c$. **C** $\mathcal{P}(t_f/t_B)$ for two different system-bath coupling strengths $\tilde{g} = 0.2\,t_B^{-1}$ [black solid (AC-QUDIT) and green dashed lines (STA)] and $\tilde{g} = 1.0\,t_B^{-1}$ [blue solid (AC-QUDIT) and red dashed lines (STA)] with $m = 0.5\,m_B$ and

$v(t_f/t_B) = 1.5\,c$. **D** Survival probability for transport through vacuum ($\tilde{g} = 0$) at non-adiabatic final speed $v(t_f/t_B) = 1.5c$, i.e., $1.5/2at_c$ ($a$ is the Morse trap-width parameter and $t_c$ the coherence time associated with nonadiabatic leakage from the trap), $t_B = 2t_c$ for simplicity and $m = 0.5\,m_B$ (can be in any arbitrary unit) obtained by AC-QUDIT (black) and with CDF (green). As before we have used the Lagrange multipliers $\lambda = 1$, $\lambda_1 = 1$. In all the panels the vertical dashed lines indicate the $t_f/t_B = t_c/t_B$ point on the time axis.

Remarkably, AC-QUDIT turns out to be advantageous compared to constant velocity transport for any finite coupling strength (see Fig. 2B).

The survival probability decreases with the increase in impurity mass $m$ (scaled to the phonon mass $m_B$) since the non-adiabaticity grows with $m^2$ ($\frac{H_{ne}}{\omega_{en}} \propto m$) (SI). Still, AC-QUDIT yields higher fidelty than constant speed transfer, for a given $m$ (see Fig. 2C).

Importantly, for appreciable system-bath coupling $\tilde{g}$, the prescribed AC-QUDIT acceleration becomes nonlinear in time. In Fig. 2D we observe that the stronger the coupling $\tilde{g}$, the larger is the deviation of the trap-acceleration from the constant value which is optimal for weak couplings. From the inset displaying the corresponding speeds we conclude that the potential travels faster through the medium as the coupling gets stronger, arriving at a longer distance within the same time (as indicated by the area below the speed curves). Since the final speed is adiabatic, the integral term in Eq. (8) can be neglected, and the displayed behavior is then due to the first two terms on the r.h.s. of this equation, where the functions $\eta$ and $\zeta$ become appreciable as the coupling grows. By contrast, for weak coupling, the constancy of acceleration implies that $\ddot{v}(t) = 0$, so that all the terms on the r.h.s. of Eq. (8) become negligible.

The survival probabilities obtained by AC-QUDIT surpass those obtained by CDF, as shown numerically in Fig. 3, in accordance with the rigorous theoretical conditions developed before (see Eq. (11), "Methods"). Since the CDF method is designed for any arbitrary trajectory, for illustration in Fig. 3 we have used the optimal trajectory

obtained from our method which minimzes both non-adiabatic and bath-induced leakages simultaneously. Any other arbitrary trajectory may result in even lower values of the survival probability under CDF in the presence of dissipation, due to non-optimal suppression of bath-induced effects.

AC-QUDIT becomes significantly advantageous compared to CDF primarily for non-adiabatic final trap speed $v(t_f) > c$ in the dissipative case (Fig. 3A). This is comparable with the advantage of AC-QUDIT relative to STA for non-adiabatic speeds in dissipationless transport (Fig. 3D). Upon increasing the impurity mass $m$ the advantage of AC-QUDIT becomes appreciable even for final speeds, $v(t_f) < c$ (Fig. 3B). The reason is that the adverse effect of acceleration-PSD and velocity-acceleration CSD incurred by CDF (Eq. (17)) scales with $m^2$.

The advantage of AC-QUDIT increases with the decrease in the dissipative system-bath coupling $\tilde{g}$, (Fig. 3C), although the CDF does not change the bath-induced transition rates $\Delta_{n\epsilon}^k$ as discussed before. This reflects the dependence of optimal trajectories in AC-QUDIT on $\tilde{g}$, which cannot be matched by their STA counterparts. Thus, we find that AC-QUDIT yields much higher survival probabilities compared to STA and constant speed transport both in the presence and in the absence of system-bath coupling.

Importantly, the CDF method is designed to work only for $v(0) = v(t_f) = 0$[27,78]. We show, by contrast, that our optimal trajectory yields significantly higher survival probabilities than CDF, even for $v(t_f) \neq 0$, proving its superiority and wide range of applicability (see SI XI, Supplementary Fig. S-2, Supplementary Fig. S-4 (B) for $v(t_f) = 0$).

## Discussion

We have put forth the general and powerful AC-QUDIT method of maximizing the fidelity of fast, non-adiabatic quantum transport in dissipative media by means of the optimal velocity-or-acceleration control Eq. (8) or its dissipationless counterpart Eq. (9). This method is universally applicable to the study of diverse topical issues, such as the evolution of atomic wavepackets[80,81], solitonic transport[82], transport of trapped atoms and ions in quantum information processing protocols[16–21], quantum refrigeration by trapped impurities in condensates[15], radiation from fast charged particles in solids[83] among others.

All such processes should be completed as fast as possible, not only to shorten their duty cycle, but also to minimize the inherently quantum hurdle of wavepacket spread and dissipation / decoherence. Moving field-induced traps (tweezers), can reduce the wavepacket leakage, but, unless the trap is very deep (which may require overly intense fields)[20], fast transport triggers non-adiabatic leakage to the continuum.

Remarkably, we have shown that our method yields transport efficiencies ≥95% at $t_f \lesssim t_c$, even for a very shallow potential trap that supports a single bound-state (the Friedrichs model), in case of transport through vacuum (Fig. 3D). Similar considerations can apply to the control of molecular dissociation and collisions[23] by engineering the potential surface on which the molecular wavepacket moves, and thereby the wavepacket speed and trajectory.

For transport through BEC, the time scales we have explored are shorter than the typical time for the emission of elementary excitations (polaron formation) in the BEC as well as their lifetimes, but longer than the time-scales on which the assumption of one-dimensionality of the BEC breaks down: the formation- and life-times of elementary excitations exceed the coherence time of impurity dynamics in BEC, $t_B$[74,84], while the violation of one-dimensionality is manifest only at times shorter than the inverse transversal trapping frequency, which correspond to elementary excitations with energies high enough to break down the one-dimensionality[85,86].

The nonlinear EL integro-differential equation (see "Methods", Eqs. (12), (13)) for the optimal wavepacket trajectory has broad applicability in the scenarios discussed above. Its linearized, analytically soluble version (8), agrees with fully-numerical solutions of the former (see SI XI, XIII, Supplementary Fig. S-1) even for fast, supersonic trajectories.

Our method presents an advantageous alternative to the conceptually powerful optimal control of non-adiabatic transport by counter-diabatic driving (alias shortcuts to adiabaticity−STA)[26–28,30–32,34,35] which faces limitations when open quantum systems are concerned (as discussed in the Introduction[51,59]). Instead, we have tackled the formidable problem of quantum transport in dissipative media by dynamical control based on a generalized Wigner−Weisskopf approach that can effectively counter both quantum dissipation and nonadiabaticity of moving wavepackets, resulting in maximal transport fidelity. This method is a paradigmatic generalization of the Kofman-Kurizki (KK) non-Markovian universal control formula for discrete quantum variables, to wavepackets[3,5,8].

Our general approach is rigorously proven to outperform STA protocols based on CDF for a broad class of transport trajectories and trapping potentials, while the invariant based STA methods are restricted only to the specific, Lewis-Leach class of potential traps[35]. Importantly, for non-adiabatic transfer, our method yields significantly higher transfer fidelity than STA.

This approach is also conceptually advantageous compared to quantum methods based on feedback[87] since it does not rely on measurements and thus can be included in quantum information processing schemes.

A central insight obtained from our approach is that non-adiabatic leakage constitutes an effective bath, that has acceleration-dependent colored spectrum, and is added to the standard environmental bath. Thus, even in the absence of a dissipative environment, i.e., transport through vacuum, there is inevitable loss of fidelity due to this effective bath which cannot be avoided by methods based on STA (see "Methods"). For non-zero system-bath coupling, we have found that it is essential to account for coupling between the trapped wavepacket and the environmental bath, beyond the Lamb-Dicke limit[13,67], in order to find an optimal trajectory that can simultaneously minimize the effect of the two baths. The survival probability (Loschmidt echo) is then determined by the spectra of both baths whose Fourier components are modulated by the time-varying trap-speed.

To sum up, this approach opens an avenue towards the control of quantum transport in dissipative media and has the potential for the discovery of unexplored features of the transport.

## Methods

### Fidelity and Loschmidt echo

Fidelity of a dynamical state is defined as its susceptility to perturbations, given by the overlap of the perturbed and unperturbed states at a given instant (Loschmidt echo[12,69]). In our problem, the perturbed state at $t > 0$ corresponds to $|\psi(t)\rangle$ which accounts for the effects of phonon-induced and non-adiabatic transitions (perturbations), while the unperturbed state at the same instant is $|\nu(t)\rangle = |n(t)\rangle \otimes |0_{bath}\rangle$ up to a global phase (without transitions induced by the perturbations).

In our problem $H_{SB}$ entangles advanced / retarded and instantaneous wavepacket states with excited or ground states of the bath, ensuring conservation of momentum. We note that the reduced density matrix of the system at $t > 0$ is a mixture of such advanced / retarded and instantaneous states (a mixed state) since the combined system-bath state is entangled. Detection of a wavepacket state advanced by momentum $k$, implies corresponding collapse of the bath to an excited state having momentum $-k$. Similarly, detection of an instantaneous wavepacket state, ensures that the bath collapses to the ground state $|0_{bath}\rangle$ i.e., the survival probability in the instantaneous bound state (fidelity) is given by the Loschmidt echo, which is here the probability of finding the combined system in the many-body state $|\nu(t)\rangle$ (see SI IV for details). Only the diagrams in Fig. 1C (i) corresponding to no quanta exchange (purely non-adiabatic) and virtual quanta exchange (purely bath-mediated) can result in the state $|\nu(t)\rangle$ at $t > 0$. The diagrams in Fig. 1C (ii) and (iii) corresponding to real quanta exchange with the bath, result in advanced / retarded system states and corresponding excited bath states. A classical analog of this scenario is that of a catcher on a moving train that throws a ball vertically upwards. To an observer on the platform (in the laboratory frame), both the catcher and the ball undergo a horizontal displacement before the falling ball lands back in the catcher's palm (in an instantaneous trap state). If instead the ball was launched up in the train along with a finite horizontal momentum relative to the train, the catcher may have to move backward or forward in order to catch the ball (in an advanced or retarded trap), thereby compensating for the additional momentum imparted. This reflects the fact that advanced / retarded states do not contribute to fidelity in the instantaneous state.

It is important to note that Loschmidt echoes are routinely measured experimentally, in studying quantum many-body dynamics, and is often used as a probe for dynamical phase transitions[12,74,88–92].

### Optimal wavepacket trajectory

In the spirit of the Wigner-Weisskopf method[8,63], we first integrate out the unbound sector of $A_l$ in Eq. (4) to obtain a system of differential equations for the bound sector (see SI V). Within the bound-sector, we again integrate out all states except $|\nu(t)\rangle$ to obtain the Loschmidt echo probability amplitude $A_\nu$ (see SI V, VI).

The resulting integro-differential non-local Euler-Lagrange (EL) equation has the following form (see SI VII):

$$\lambda \ddot{x}_\circ(t) = \int_0^{t_f} dt' \left\{ \dot{x}_\circ(t') \int d\epsilon \, \frac{|\mu_{n\epsilon}|^2}{\omega_{\epsilon n}} \sin[\omega_{\epsilon n}(t-t')] \right. \tag{12}$$
$$\left. - K(t,t') \right\};$$

$$K(t,t') = \frac{L}{2\pi} \int dk \, d\epsilon \, k \, |\widetilde{g}_{n\epsilon}^{(k)}|^2 \, \sin\left[ \int_{t'}^t dt'' \{\omega_{\epsilon n} + \Omega_k \right. \tag{13}$$
$$\left. + k \, \dot{x}_\circ(t'')\} \right].$$

The kernel $K(t,t')$ is expanded as a weighted sum of Fourier harmonics with frequencies $(\omega_{\epsilon n} + \Omega_k)$ and is frequency modulated (FM) via the control function $k\dot{x}_\circ(t)$[77]. This hitherto unattempted FM control results from the nonlinearity in $kx_\circ$ of the system-bath coupling Hamiltonian, when $e^{-ikx}$ is beyond the Lamb-Dicke limit in Eq. (2). This $kx_\circ$-nonlinearity is essential for controlling the bath-spectrum by changing the trap speed $v(t) = \dot{x}_\circ(t)$. If we were restricted to the linear-coupling (Lamb-Dicke) regime where $e^{-ikx} \rightarrow -ikx = -ik[x-x_\circ(t)] - ikx_\circ(t)$, then $x_\circ(t)$ would act as an external field in the instantaneous system basis, which cannot serve as a handle on the bath-induced transitions between the instantaneous levels.

By time-differentiating both sides of Eq. (12) we convert the kernel Eq. (13) into a sum of amplitude- and frequency-modulated (AFM) terms of the form

$$\dot{K}(t,t') = \frac{L}{2\pi} \int dk \int d\epsilon \, k \, |\widetilde{g}_{n\epsilon}^{(k)}|^2 \, [(\omega_{\epsilon n} + \Omega_k) + k \, \dot{x}_\circ(t)]$$
$$\times \cos\left[ (\omega_{\epsilon n} + \Omega_k) \int_{t'}^t dt'' \left\{ 1 + \frac{k}{(\omega_{\epsilon n} + \Omega_k)} \dot{x}_\circ(t'') \right\} \right]. \tag{14}$$

In the speed range $|v(t)| = |\dot{x}_\circ(t)| < v_s = |(\omega_{\epsilon n} + \Omega_k)/k|$, we obtain to the lowest order in $|kv(t)/(\omega_{\epsilon n} + \Omega_k)|$, the linearized Eq. (8). The boundary value problem (BVP) associated with Eq. (8) is solved by first reducing it to a Fredholm integral equation using a Green's function method[93,94] (see SI X) and then applying the Liouville-Neumann expansion to solve the resulting integral equation[95]. Numerically, we solve the fully non-linear EL equation by successive iterations (see SI XIII, Supplementary Fig. S-1.) and verify that the solutions match well with the results obtained analytically from Eq. (8). The corresponding results presented in Figs. 2 and 3 satisfy the condition $|v(t)| < v_s$ (see SI XI, Supplementary Fig. S-3).

## A wavepacket trapped in Morse potential and dissipated as in the Frölich model

In the Frölich model of an impurity interacting with a bath of bosons forming a BEC[12,13,66] confined in a volume $\mathcal{V}$, the parameters in Eq. (8) are $g_k = g_{-k} = g_k^* = g_{SB}\sqrt{\frac{n_0}{\mathcal{V}}}\left[\frac{(\xi k)^2}{(\xi k)^2 + 2}\right]^{\frac{1}{4}}$, $\Omega_k = c|k|\sqrt{1 + \frac{1}{2}(\xi k)^2}$. Here $\xi = 1/\sqrt{2g_B m_B n_0}$ represents the coherence length of phonons, $c = 1/(\sqrt{2}m_B \xi)$ is the BEC speed of sound, $m_B$ is the phonon mass and $n_0$ is the ground state density of the BEC, while $g_{SB}$ and $g_B$ denote the impurity-boson and inter-boson interaction strengths, respectively[13]. In a 1-d confinement of the BEC, the volume $\mathcal{V}$ is replaced by the confining length $L$. The Morse potential trap is given by $V[x-x_\circ(t)] = D\left[e^{-2a\{x-x_\circ(t)\}} - 2e^{-a\{x-x_\circ(t)\}}\right]$ (SI I). The expressions for

the transition matrix elements $\mu_{n\epsilon}$, $\Delta_{n\epsilon}^k(t)$ and $D_{n\epsilon}$, for this choice of the trapping potential, are obtained from literature (see SI II, VIII, IX).

**Relevant time-scales.** The two distinct leakage terms in Eq. (6) are associated with two different time-scales. The shortest correlation time of the continuum $t_c$ for non-adiabatic transitions is the inverse of trap-depth[10], $t_c = 1/D$. The coherence time-scale for impurity dynamics in BEC, $t_B$ is determined by the Bogoliubov coherence length $\xi$, $t_B = \xi/\sqrt{2}c$[13,74].

**Choice of units and parameters.** We express all length-scales in units of $\xi$, time-scales in units of $t_B$ and mass in units of $m_B$. The coupling strength $\widetilde{g}$ is expressed in units of $1/t_B$. We choose $\hbar = 1$, $c = 1$ and $m_B = 1$ for our numerical calculations (see Table 1). In our analysis, we set the Lagrange multiplier to be $\lambda = 1$.

In order to have a single bound-state (Friedrichs model), we must have $\sqrt{\frac{2mD}{a^2}} - \frac{1}{2} < 1$ (see SI II). In the scaled parameters, this implies $D' < \frac{9a'^2}{8m'}$, using the fact that $t_B/\xi^2 = 1$ in our units.

In order to illustrate our method, we explore the regime $t_c < t_B$ i.e., $t_c' = 1/D' < 1$ or $D' > 1$, so that $1 < D' < \frac{9a'^2}{8m'}$. In the plots of Figs. 2 and 3, we have used $D' = 2$, $a' = 1$, $m'$ i) 0.25, ii) 0.5 and $D' = 2$, $a' = 2$, $m' = 2$. Hence $t_c' = 1/D' = 0.5$. In all our computations, we choose the cut-off values of $\epsilon$ and $k'$ as $\epsilon_{max} = 5$ and $|k'|_{max} = 5$. Without loss of generality we choose $\widetilde{g} = t_B g_{SB} \sqrt{\frac{n_0}{2L}}$.

## Nonadiabaticity condition

For a closed quantum system, the adiabatic approximation is valid in the limit[96]

$$\max_{0 \le t \le t_f} \left| \frac{\langle n(t)| \frac{\partial H_s(t)}{\partial t} |\epsilon(t)\rangle}{\omega_{\epsilon n}} \right| \ll \min_{0 \le t \le t_f} |\omega_{\epsilon n}|. \tag{15}$$

For a moving potential trap (without a bath), the condition (15) becomes, in terms of the non-adiabatic transition matrix elemts $\frac{\mu_{n\epsilon}}{\omega_{\epsilon n}}$

$$\max_{0 \le t \le t_f} \left| v(t) \frac{\mu_{n\epsilon}}{\omega_{\epsilon n}} \right| \ll \min_{0 \le t \le t_f} |\omega_{\epsilon n}|. \tag{16}$$

For $m = 0.5 \, m_B$ the minimum value of $\omega_{\epsilon n}$ is $\sim 0.84 \, t_B^{-1}$. For $v(t_f) = 0.5c$ (subsonic target speed) the l.h.s. of the above inequality (16) is $0.26 \, t_B^{-1}$ which is $\ll \min |\omega_{\epsilon n}|$ and hence corresponds to an adiabatic transport. On the other hand for $v(t_f) = 1.5 \, c$ (supersonic target speed) the l.h.s. of (16) is $1.65 \, t_B^{-1}$ which is $> \min |\omega_{\epsilon n}|$ corresponding to non-adiabatic transport.

## Condition for superiority of AC-QUDIT over the CDF method

With the introduction of a CDF, the non-adiabatic transition amplitudes get modified to $\Gamma_{n\epsilon}(t)$ (see Eq. (10)). We then have:

$$\int_0^{t_f} dt_1 \int_0^{t_f} dt_2 \, \Gamma_{n\epsilon}(t_1)\Gamma_{n\epsilon}^*(t_2) = \underbrace{\left|\frac{\mu_{n\epsilon}}{\omega_{\epsilon n}}\right|^2 |v_{t_f}(\omega_{\epsilon n})|^2}_{\text{velocity-PSD}}$$
$$+ \underbrace{m^2 \int d\epsilon |D_{n\epsilon}|^2 |\dot{v}_{t_f}(\omega_{\epsilon n})|^2}_{\text{acceleration-PSD}}$$
$$+ \underbrace{2 \int d\epsilon \, \text{Re}\left[ i \, m \left(\frac{\mu_{n\epsilon} D_{n\epsilon}^*}{\omega_{\epsilon n}}\right) c_{t_f}(\omega_{\epsilon n}) \right]}_{\text{velocity-acceleration-CSD}}, \tag{17}$$

where $|\dot{v}_{t_f}(\omega_{\epsilon n})|^2$ is the power-spectral density (PSD) of the trap acceleration and $c_{t_f}(\omega_{\epsilon n}) = \int_0^{t_f} dt_1 e^{-i\omega_{\epsilon n}t_1} v(t_1) \int_0^{t_f} dt_2 \left[e^{-i\omega_{\epsilon n}t_2} \dot{v}(t_2)\right]^*$ is the complex amplitude of the cross-spectral density (CSD) between

the trap speed and the acceleration of the trap. In order to satisfy Eq. (11) we must have

$$2\,\mathrm{Re}\left[i\,m\left(\frac{\mu_{n\epsilon}D_{n\epsilon}^*}{\omega_{\epsilon n}}\right)c_{t_f}(\omega_{\epsilon n})\right] + m^2|D_{n\epsilon}|^2|\dot{v}_{t_f}(\omega_{\epsilon n})|^2 > 0 \qquad (18)$$

which holds for the entire parameter regime that we have explored.

### Inadequacy of CDF to avoid irreversible loss of fidelity

Unstable bound-state wave-packets undergo irreversible leakage to the continuum[8-10,79] as described by Eqs. (5) and (6). The exponent on the r.h.s. of Eq. (5), $J[x_\circ, \dot{x}_\circ]$, depends on the time-integral over the entire trajectory as shown in Eq. (6). The non-adiabatic contribution to $J[x_\circ, \dot{x}_\circ]$ (the first term on the r.h.s. of Eq. (6)), transforms to Eq. (17) in the presence of a CDF (STA), which is essentially $|\int_0^{t_f} \Gamma_{n\epsilon}(s)ds|^2 \geq 0$.

Likewise, the phonon-mediated term is $|\int_0^{t_f} \frac{L}{2\pi} \int dk \, \Delta_{n\epsilon}^k(s)ds|^2 \geq 0$. Since the coupling $H_{SB}$ depends on the canonical position of the trapped-particle $x$, taking the trap velocity $v(t_f) = 0$ and trap-acceleration $\dot{v}(t_f) = 0$ at $t = t_f$ is not sufficient to cancel the phonon contribution to the irreversible loss of fidelity. This shows that irreversible loss of fidelity cannot be avoided via CDF. At best one might strive for $|\int_0^{t_f} \Gamma_{n\epsilon}(s)ds| = 0$ by this method. Yet Eq. (10) suggests that this would require $\gamma_{n\epsilon}(t) = im\dot{v}(t)D_{n\epsilon}e^{-i\omega_{\epsilon n}(t)}$ for all $t > 0$ and $\epsilon > 0$. This in turn requires that $v(t)\frac{\mu_{n\epsilon}}{\omega_{\epsilon n}}e^{-i\omega_{\epsilon n}t} = im\dot{v}(t)D_{n\epsilon}e^{-i\omega_{\epsilon n}t}$, i.e., $v(t) = v(0)\exp\left[-i\int_0^t ds\left(\frac{\frac{\mu_{n\epsilon}}{\omega_{\epsilon n}}e^{-i\omega_{\epsilon n}s}}{mD_{n\epsilon}e^{-i\omega_{\epsilon n}s}}\right)\right]$ for all $t > 0$ and $\epsilon > 0$. It is however impossible for a single velocity profile to satisfy an infinite set of equations of the form shown above for each value of $\epsilon > 0$, $t > 0$.

This proves the fact that the irreversible loss of fidelity cannot be avoided by merely setting $v(t_f) = 0$ and $\dot{v}(t_f) = 0$ at the end point $t = t_f$.

## Data availability

The source data supporting the numerical findings of this study have been deposited in Github at[97].

## Code availability

The codes supporting the findings of this study are available from the corresponding author upon request.

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

## Acknowledgements
G.K. is supported by the US-Israel NSF-BSF Program and DFG (FOR 2724). IM is supported by the grant SFB F65 "Taming Complexity in PDE system" by the Austrian Science Fund (FWF), the FQXI program on "Fueling quantum field machines with information" and the Wiener Wissenschafts- und Technologie-Fonds (WWTF) project No MA16-066 (SEQUEX). I.M. acknowledges support from the European Research Council via ERC/AdG "Emergence in Quantum Physics" (EmQ) under Grant Agreement No. 101097858 and the Austrian Science Fund (FWF) via SFB F65 "Taming complexity in PDE systems". X.C. acknowledges the National Natural Science Foundation of China (Grants Nos. 12075145), the project grant PID2021-126273NB-I00 funded by MCIN/AEI/10.13039/501100011033 and by "ERDF A way of making Europe" and "ERDF Invest in your Future", the Spanish Ministry of Economic Affairs and Digital Transformation through the QUANTUM ENIA project call-Quantum Spain project, the Basque Government through Grant No. IT1470-22, and the Severo Ochoa Centres of Excellence program through Grant CEX2024-001445-S. A.C. thanks Mayank Shreshtha for designing Fig. 1. B.A. acknowledges the support, encouragement and assistance of Hillel Aharoni. A.C. and B.A. thank Pritam Chattopadhyay, Saikat Sur and Ankita Ganguly for discussions.

## Author contributions
G.K. with I.M. and X.C. have conceived and formulated the problem. A.C. developed the theory, and A.C. and B.A. developed the numerical methods. A.C. and B.A. obtained the analytical and numerical solutions and identified the noteworthy physical results. All authors have jointly written the paper.

## Competing interests
The authors declare no competing interests.
