## [Transparent Peer Review file · Nature Communications]

Quantum Transport Protected by Acceleration From Nonadiabaticity and Dissipation

Corresponding Author: Dr Arnab Chakrabarti

Version 0:

Reviewer comments:

Reviewer #1

(Remarks to the Author)

This paper describes an algorithm to control open driven quantum systems. Its results are relevant for applications. I have some doubts about the realisability of the proposed trajectory control, but this could be better explained by the authors, together with some minor issues that I ask them to address:

- 1) II. 47-48: counterdiabatic methods have been applied to non-integrable problems as well, see e.g. <https://arxiv.org/abs/2406.17545>. So maybe here I would be a bit more careful with such generalising statements.
- 2) Sec. II D, starting from I. 239: extensions to open quantum systems of STA approaches are discussed in the literature. Maybe, a direct comparison can be given, or at least a partial discussion of such approaches with respect to the presented method here. See, e.g., <https://www.nature.com/articles/s41598-019-39731-z>, Scientific Reports 9, 4048 (2019).
- 3) I was wondering whether the results of figs. 2 and 3 could not be better presented. Maybe including the data shown in more elaborated plots, 3D?
- 4) The ref. section:
 - i) I have found at least 13 self citations! Probably not all justified.
 - ii) another method of protecting wave packet evolution would be non-dispersive wave packet control, see a Phys. Rep. by Delande and Zakrzewski from 2002. Maybe worth mentioning in this context.
 - iii) next to ref.s 9-11, there is book edited by Schlagheck and Keshavamurthy: Dynamical Tunneling Theory and Experiment, 2011, containing similar material.

Reviewer #2

(Remarks to the Author)

The paper "Quantum Transport Protected by Acceleration from Nonadiabaticity and Dissipation" introduces the Acceleration-Controlled Quantum Dissipative Transport (AC-QUDIT) method to achieve high-fidelity quantum wavepacket transport in dissipative environments, even for shallow trapping potentials. By optimizing the survival probability of the initial state, in the presence of non-adiabatic transitions and dissipation by the bath, the authors derive an integrodifferential equation for the optimal trap speed. Then, they compare the AC-QUDIT protocol with the constant speed protocol and a Shortcut to Adiabaticity protocol, for the optimal transfer of an impurity through a 1D BEC. The numerical results demonstrate the superiority of AC-QUDIT for different values of the system parameters like final speed, transfer time, impurity mass and coupling with the bath.

The paper is well written and the AC-QUDIT method is sufficiently described, while it is thoroughly explained in the supplementary material. The proposed transport method appears to have a significant advantage over state-of-the-art

methods like STA and the article convincingly demonstrates its superior performance in achieving high-fidelity quantum wavepacket transport in dissipative environments, making it a promising tool for future experimental applications in diverse research areas. For these reasons we believe that the article warrants publication in Nature Communications, after the authors consider the following rather minor points:

1. In Fig. 2D we observe that for larger coupling constant the acceleration substantially deviated from the constant value, which appears to be optimal for smaller couplings. From the inset displaying the corresponding speeds we conclude that the potential travels faster through the medium, arriving at a longer distance in the same time (area below the speed curves). Since the final speed is adiabatic, the integral term in Eq. (7) can be neglected, and the observed behavior is due to the first two term on the rhs of the equation, where functions η and ζ become appreciable for strong coupling. On the other hand, for the weak coupling case, the constancy of acceleration implies that $v''=0$, so all the terms of the rhs of Eq. (7) appear to be negligible. The authors may would like to add the corresponding comment.

2. In the caption discussing Fig. 2A, the citation of blue and green colors should be interchanged.

3. The authors may would like to cite the following related work in the introduction, where they discuss STA methods:

X. Chen, E. Torrontegui, D. Stefanatos, J.-S. Li, and J. G. Muga, Optimal trajectories for efficient atomic transport without final excitation, Physical Review 84, 043415 (2011).

4. Line 161, the states r, s are not explained.

5. Page 6, line 345, "dissipationless".

Reviewer #3

(Remarks to the Author)

Reviewer #4

(Remarks to the Author)

Report on: "Quantum Transport Protected by Acceleration From Nonadiabaticity and Dissipation" by Chakrabarti, et al.

This paper considers fast transport of a wavepacket which is confined in a potential trap subject to a non-Markovian and Gaussian bath. The authors develop a framework to optimize the fidelity of finding the system-bath state to be the initial bound state (at time t) with no bath excitations. By taking leading order in \dot{x}_o and H_{SB} , the authors obtain an analytical expression of the fidelity in terms of transition amplitude-like quantities induced by non-adiabatic (system unitary) dynamics and bath-induced dynamics. Furthermore, the authors give an approximate analytical expression (linearized integro-differential EL equation) that allows optimizing the fidelity. The authors compare their results with the widely studied STA technique, in particular, the CDF approach designed for isolated quantum systems, and find that their method (AC-QUDIT method) outperforms CDF.

Developing a control method that optimizes the fidelity of finding the system to be the target state is important in experiments, including transfer of atoms and ions trapped in a potential (e.g. optical tweezers) through noisy environments. Therefore, STA and other control techniques, such as the dynamical decoupling technique, have been extended to open quantum systems. In this line, the results obtained by the authors contribute to the development of quantum control techniques by providing a method that allows optimization of the control parameter (e.g. the speed of moving the trapping potential). However, the framework relies on the lowest order in the speed of the potential and the system-bath coupling strength, which seems to be basically a linear-response relation. Therefore, the validity of their methods, in particular to the case of fast non-adiabatic transport claimed in the abstract, is unclear. There are several settings (see comments (1) and (2)) that seem unnatural in experimental situations and require physical justification or modification of the results. The presentation is quite technical, and the presentation could be further improved to increase the accessibility to general readers. These points may reduce the significance of the paper, and my general impression is that the paper is more suitable for specialized journals.

Detailed comments:

(1) As far as I understand, the fidelity (5a) is obtained by comparing the system-bath quantum state at time t : $|\psi_{SB}(t_f)\rangle = U_{SB}(t_f, 0)|\psi_{SB}(0)\rangle$ and the final target state $|\nu(t_f)\rangle = |\nu(t_f)\rangle \otimes |0_{bath}\rangle$, where $U_{SB}(t_f, 0)$ is the unitary operator by solving the Schrodinger equation. Because of this choice, only the case (i) in Fig. 1. C (i.e., purely non-adiabatic and no quanta exchange processes) contributes to the fidelity. However, the quantity of interest should be the fidelity of the reduced density matrix of the system $\rho_S(t_f) = \text{Tr}_B[\rho_{SB}(t_f)]$ to be in the target state $|\nu(t_f)\rangle$, defined by $\langle \nu(t_f) | \rho_S(t_f) | \nu(t_f) \rangle$, because this quantity expresses the probability of finding the system in the bound state of the trapping potential.

The bath degrees of freedom is typically not accessible in experiments, and therefore the physical meaning of calculating the fidelity to find both the system in the target bound state $|n(t)\rangle$ and also the bath to be in the ground state $|0_{\text{bath}}\rangle$ is not clear. I think the results should be reformulated by using the fidelity by using the reduced density matrix of the system.

(2) In line 217, I guess the authors mean that Eq. (7) is solved under the boundary condition $v(0)=0$ and $v(t_f)=\text{some fixed value}$ (c.f. 0.5c, 1.5c). I think fixing the initial position and the trap-speed sounds reasonable in experimental situations. However, the choice of fixing the final trap-speed is unclear and needs physical justification. I would assume that fixing the final center position of the trap $x_o(t_f)$ is more natural and relevant to experimental situations.

(3) As the authors write in the supplementary material, the fidelities (survival probabilities) P are plotted using the analytical expressions Eqs. (28) and (29) [Eqs. (5a) and (5b) in the main text], which are valid in the leading order in \dot{x}_o and H_{SB} . In particular, if I understand correctly, Eq. (5a) should read $P(t_f)=1 - J[x_o, \dot{x}_o] + \text{higher-order terms}$, which is then approximated by $P(t_f)=\exp(-J[x_o, \dot{x}_o])$. If this is true, then by noting that, $\exp(-J)$ and $1-J$ start to deviate from $O(J^2)$, for example, when P is below 0.9. Therefore, I suspect that some part of plots in Fig. 2 and Fig. 3 are outside the validity of the analytical expressions (5a) and (5b) (for example, AC-QUDIT methods: black solid curves with $0.4 < t_f/t_B$ in Fig. 3B, and Fig. 3C, STA methods: most of them in Fig. 3).

(4) The authors write in line 271-273 that “By contrast, $\delta^k \epsilon_n(t)$ remains invariant since CDF does not involve bath-induced transitions” but I think this is correct only if terms proportional to $g_k^2 \dot{v}^2$ are neglected.

(5) The authors write in line 406 that “Our general approach is rigorously proven to outperform STA protocols based on CDF for a broad class of transport trajectories and trapping potentials” but I think this sentence is misleading. I think the authors should write that up to leading orders in \dot{x}_o and H_{SB} , a condition to outperform CDF has been derived. Related to the comment (2), the authors choose to fix the final trap-speed to be non-zero in all numerical calculations. This choice obviously makes CDF insufficient, even in the isolated case as plotted in Fig. 3D. Therefore, one might suspect that the comparison between AC-QUDIT and CDF discussed in this paper is limited to specific conditions in which CDF is insufficient. If the final center position of the trap is fixed, and the final trap velocity can be freely chosen, I guess that the plot would become very different, since one could choose $\dot{x}_o(t_f)=0$ to make the CDF reproduce the adiabatic state at final times (note that for $0 < t < t_f$, the system does not necessarily follow the adiabatic state).

(6) Does the condition given in line 189 satisfied for the parameters used in Fig. 2, 3? Please add this discussion somewhere.

(7) From line 263 to 266: The authors write “This irreversible loss of fidelity cannot be avoided by merely making the velocity and acceleration vanish at the boundary points, as is usually done to achieve transition-less transport via STA [27,47].” Do the references [27,47] show that “This irreversible loss of fidelity cannot be avoided”, or they present the usual STA by “making the velocity and acceleration vanish at the boundary points”? If it is the latter case, please cite a paper or add additional discussions (numerical calculations) to clarify this statement.

(8) Suggested references.

System-bath control methods for non-Markovian systems:

<https://doi.org/10.1103/PhysRevX.10.031015>

Studies deriving exact analytical expressions and bounds for the fidelity for non-Markovian dynamics:

<https://doi.org/10.1103/PhysRevB.75.214308>

<https://doi.org/10.1103/PhysRevLett.97.200404>

<https://doi.org/10.1103/PhysRevLett.127.150401>

Typos:

Line 126: t should be replaced by 0

Line 161: $\langle r(t) \rangle$ and $|s(t)\rangle$ should be replaced by $\langle n(t) \rangle$ and $|\epsilon(t)\rangle$.

Fig.2A caption: (green) and (blue) are opposite. They do not match the figure. In addition, $v(t_f/t_B)$ should be replaced by $v(t_f)$. Similar typo for $P(t_f/t_B)$.

Version 1:

Reviewer comments:

Reviewer #1

(Remarks to the Author)

I am happy with the corrections made following my suggestions. Also the replies to the other referee, in particular, reviewer No. 4, seem satisfactory. Some of the newly cited references have appeared in print now, so please cite the "originals", e.g. 37 and 60

Phys. Rev. A 111, L031301 (2025) and Phys. Rev. B 111, 064301 (2025). Moreover, some refs. have titles, other not, so please use a uniform presentation.

With those minor changes the paper may be published.

Reviewer #2

(Remarks to the Author)

The authors have addressed our observations in the revision, we thus recommend the publication of the article in Nature Communications.

Reviewer #3

(Remarks to the Author)

Reviewer #4

(Remarks to the Author)

I have read the revised manuscript and the response. The authors have adequately responded to most of my comments, and the clarity of the manuscript has been improved (see "Other minor points").

Unfortunately, there is one point (the applicable regime of the proposed method) which I am not fully convinced. If the authors can clarify this point with adequate revision of the manuscript, the manuscript may be suitable for publication in Nature communications.

While I do understand that the expression for the fidelity is obtained by resummation of all second-order self-energy diagrams mentioned by the authors (meaning that the step from Eq. (1) to Eq. (2) in the response letter is exact), the starting point of Eq. (1) or (S28) requires "retaining only the lowest order terms in the trap-speed and system-bath coupling" as mentioned by the authors. Therefore, I am not fully convinced about the applicable regime of the expression (5a) and (5b) in the main text. For example, the authors write in the reply that "This allows our method to be valid for fast non-adiabatic transport and strong system-bath couplings, as long as the relaxation is exponential", and "the condition we derive is not restricted to leading orders in" trap-speed and system-bath coupling, but the logic of these statements were unclear to me. Please explain why does the assumption of taking lowest order in the trap-speed and system-bath coupling to derive Eq. (1) is consistent with the above statement.

As far as I can see from Eq. (5b), the quantities depend only on the system energy eigen-energies, eigenstates (and their derivatives), system-bath coupling operator, and the bath parameters, and therefore the information of the generically nonequilibrium time-evolution of the system-bath state is not captured. I guess this is why the authors can optimize the protocol by solving the EL equation, but on the other hand, this expression resembles me of linear-response relation where one could express quantities based only on adiabatic (e.g., energy eigenstates, etc) quantities and that optimization method for Hamiltonian parameters are possible.

If there are some numerical plots obtained from numerical solution of the exact system-bath dynamics or using some non-Markovian numerical simulation methods for Gaussian baths, the validity of the methods proposed in this manuscript would be clearer (at least, the case of vanishing system-bath coupling strength should be doable).

Other minor points:

About the choice of the fidelity, I agree that if one could access all degrees of freedom, including the bath state, then the fidelity mentioned in the main text could be measured (as explained in the reply letter in terms of Loschmidt echo or similar methods). The authors have expanded physical discussions, including the scattered wavepacket discussion, which may clarify why the authors focus on the fidelity between final state and the instantaneous bound state with no bath excitations. About the choice of the boundary conditions, I agree that fixing two values would determine the whole quantities. I guess the following point clarifies the choice "In order to show the superiority of our method over CDF, we have chosen the fixed final trap speed to be non-zero, a case where CDF fails but our method works well". I also wanted to understand how the plots Fig. 2 and 3 look like when the final position of the trap is fixed, but this may be optional.

The choice of vanishing final trap speed and comparison to CDF method is now much clearer, and I agree with the authors that their method is quite effective compared to the CDF method.

Version 2:

Reviewer comments:

Reviewer #4

(Remarks to the Author)

The authors have addressed all points raised in the previous report, as far as possible. I think the revised manuscript is now suitable for publication in Nature communications.

Response to Reviewers' Comments

REVIEWER #1

Comment:

This paper describes an algorithm to control open driven quantum systems. Its results are relevant for applications. I have some doubts about the realisability of the proposed trajectory control, but this could be better explained by the authors, together with some minor issues that I ask them to address:

Response:

We thank the reviewer for finding our work relevant for applications. We have addressed all the issues raised by the reviewer in the revised version as detailed in the response below. We hope that the revised manuscript is found suitable for publication in Nature Communications.

Comment:

1) II. 47-48: counterdiabatic methods have been applied to non-integrable problems as well, see e.g. <https://arxiv.org/abs/2406.17545>. So maybe here I would be a bit more careful with such generalising statements.

Response:

We thank the reviewer for this useful suggestion. We have rectified this statement to:

“In general, STA is mainly geared to closed, stable quantum systems [35]...”

We have expanded the discussion of this issue, as detailed in the response to Comment 2 (see below). We have also included the reference <https://arxiv.org/abs/2406.17545> in the revised manuscript.

Comment:

2) Sec. II D, starting from l. 239: extensions to open quantum systems of STA approaches are discussed in the literature. Maybe, a direct comparison can be given, or at least a partial discussion of such approaches with respect to the presented method here. See, e.g., <https://www.nature.com/articles/s41598-019-39731-z>, Scientific Reports 9, 4048 (2019).

Response:

Following the reviewer's suggestion, we have now included a short discussion of the current extensions of STA to open quantum systems. We now write in the Introduction:

“In general, STA is mainly geared to closed, stable quantum systems [35] since, being a hamiltonian method, it is apriori unclear, to what extent can STA suppress irreversible bath effects [35, 40, 41] ? One should be mindful that the success of any Hamiltonian protocol for a lossy quantum system depends on the typical ratio of the level-width to level-spacing. Yet, there have been numerous extensions of STA techniques to open quantum systems, mostly of either discrete variables or harmonic potentials (invariants quadratic in momentum) [42 – 62]. Application of the invariant based STA in open quantum systems, as in [42 – 50] has limited applicability for arbitrary (non-harmonic) trapping potentials. Extensions of the CDF method to transitionless driving of open quantum systems, typically composed

of discrete variables, may involve non-Hamiltonian or non-Hermitian control [51, 59], as opposed to the unitary control for closed-system STA.

On top of the difficulties in using STA for general open quantum systems, none of the above methods can simultaneously control the irreversible leakage of an unstable wavepacket due to combined non-adiabatic and bath-induced transitions in a realistic model, where the time-dependence of the system Hamiltonian induces changes in the bath-induced leakage. This is the challenging problem we address in this work.”

We have included the suggested reference in this discussion.

3) *I was wondering whether the results of figs. 2 and 3 could not be better presented. Maybe including the data shown in more elaborated plots, 3D?*

Response:

Unfortunately, we could not find a way to present the results in 3D plots while conveying all the relevant information.

Comment:

4) *The ref. section:*

i) *I have found at least 13 self citations! Probably not all justified.*

Response:

Following the reviewer’s suggestion, we have reduced the number of self citations in the revised manuscript. Specifically we have removed references i) Proc. Nat. Acad. Sci., 112, 3866 – 3873, (2015), ii) Nature, 405, 546, (2000) and iii) J. Phys. B. At. Mol. Opt. Phys., 49, 125503, (2016).

Comment:

ii) *another method of protecting wave packet evolution would be non-dispersive wave packet control, see a Phys. Rep. by Delande and Zakrzewski from 2002. Maybe worth mentioning in this context.*

Response: We thank the reviewer for this important reference, which we have now included.

We now write in the Introduction:

“The spread of stable quantum wavepackets can be suppressed by a resonant drive [11]. However, this method does not apply to the scenarios discussed here.”

Comment:

iii) *next to ref.s 9-11, there is book edited by Schlagheck and Keshavamurthy: Dynamical Tunneling Theory and Experiment, 2011, containing similar material.*

Response:

Following the reviewer’s suggestion we have included this reference in the revised manuscript.

REVIEWER #2

Comment:

The paper "Quantum Transport Protected by Acceleration from Nonadiabaticity and Dissipation" introduces the Acceleration-Controlled Quantum Dissipative Transport (AC-QUDIT) method to achieve high-fidelity quantum wavepacket transport in dissipative environments, even for shallow trapping potentials. By optimizing the survival probability of the initial state, in the presence of non-adiabatic transitions and dissipation by the bath, the authors derive an integrodifferential equation for the optimal trap speed. Then, they compare the AC-QUDIT protocol with the constant speed protocol and a Shortcut to Adiabaticity protocol, for the optimal transfer of an impurity through a 1D BEC. The numerical results demonstrate the superiority of AC-QUDIT for different values of the system parameters like final speed, transfer time, impurity mass and coupling with the bath.

Response:

We thank the reviewer for accurately summarizing our work.

Comment:

The paper is well written and the AC-QUDIT method is sufficiently described, while it is thoroughly explained in the supplementary material. The proposed transport method appears to have a significant advantage over state-of-the-art methods like STA and the article convincingly demonstrates its superior performance in achieving high-fidelity quantum wavepacket transport in dissipative environments, making it a promising tool for future experimental applications in diverse research areas. For these reasons we believe that the article warrants publication in Nature Communications, after the authors consider the following rather minor points:

Response:

We are grateful that the reviewer has found our manuscript well written and the work suitable for publication in Nature Communications after addressing the minor points raised. We have addressed all these minor points in the revised version of the manuscript.

Comment:

1. In Fig. 2D we observe that for larger coupling constant the acceleration substantially deviated from the constant value, which appears to be optimal for smaller couplings. From the inset displaying the corresponding speeds we conclude that the potential travels faster through the medium, arriving at a longer distance in the same time (area below the speed curves). Since the final speed is adiabatic, the integral term in Eq. (7) can be neglected, and the observed behavior is due to the first two term on the rhs of the equation, where functions η and ζ become appreciable for strong coupling. On the other hand, for the weak coupling case, the constancy of acceleration implies that $v'' = 0$, so all the terms of the rhs of Eq. (7) appear to be negligible. The authors may would like to add the corresponding comment.

Response:

We thank the reviewer for this helpful suggestion. We have addressed the above comment in the revised version where we now write:

"In Fig. 2D we observe that the stronger the coupling \tilde{g} , the larger is the deviation of

the trap-acceleration from the constant value which is optimal for weak couplings. From the inset displaying the corresponding speeds we conclude that the potential travels faster through the medium as the coupling gets stronger, arriving at a longer distance within the same time (as indicated by the area below the speed curves). Since the final speed is adiabatic, the integral term in Eq. (7) can be neglected, and the displayed behavior is then due to the first two terms on the r.h.s. of this equation, where the functions η and ζ become appreciable as the coupling grows. By contrast, for weak coupling, the constancy of acceleration implies that $\ddot{v}(t) = 0$, so that all the terms on the r.h.s. of Eq. (7) become negligible. ”

Comment:

2. In the caption discussing Fig. 2A, the citation of blue and green colors should be interchanged.

Response:

We again thank the reviewer for carefully reading our manuscript. We have made the necessary correction in the revised version.

Comment:

3. The authors may would like to cite the following related work in the introduction, where they discuss STA methods: X. Chen, E. Torrontegui, D. Stefanatos, J.-S. Li, and J. G. Muga, *Optimal trajectories for efficient atomic transport without final excitation*, *Physical Review* 84, 043415 (2011).

Response:

We have added the suggested reference in the revised version.

Comment:

4. Line 161, the states r , s are not explained.

Response:

We thank the reviewer for pointing this out. We have made the necessary corrections in the revised version.

Comment:

5. Page 6, line 345, “dissipationless”.

We have corrected this typo. We thank the reviewer for pointing this out.

REVIEWER #3

Comment:

Response:

We thank reviewers #3 and #4 for their detailed critique. We have addressed all the concerns jointly raised by reviewers #3 and #4 in the revised version, as described below.

REVIEWER #4

Comment:

This paper considers fast transport of a wavepacket which is confined in a potential trap subject to a non-Markovian and Gaussian bath. The authors develop a framework to optimize the fidelity of finding the system-bath state to be the initial bound state (at time t) with no bath excitations. By taking leading order in \dot{x}_o and H_{SB} , the authors obtain an analytical expression of the fidelity in terms of transition amplitude-like quantities induced by non-adiabatic (system unitary) dynamics and bath-induced dynamics. Furthermore, the authors give an approximate analytical expression (linearized integro-differential EL equation) that allows optimizing the fidelity. The authors compare their results with the widely studied STA technique, in particular, the CDF approach designed for isolated quantum systems, and find that their method (AC-QUDIT method) outperforms CDF.

Response:

We thank the reviewers for summarizing our work. Our method uses the Wigner-Weisskopf theory which, as detailed below, results in a resummation of all second-order diagrams (second order in \dot{x}_o and H_{SB}) in the expression for the survival probability. It is a non-perturbative theory that goes beyond the Fermi-Golden Rule expressions for transition rates (see response to detailed comments below) and hence, is not a linear response theory.

We have derived the exact EL equation for the optimal trap trajectory and solved it numerically by the method of successive approximations (Picard iterations), the details of which were provided in the SI. We have also shown that the solution of the approximate (linearized) version of the EL equation in the main text, matches well with the exact numerical solution (see SI Fig. S-1).

Comment:

Developing a control method that optimizes the fidelity of finding the system to be the target state is important in experiments, including transfer of atoms and ions trapped in a potential (e.g. optical tweezers) through noisy environments. Therefore, STA and other control techniques, such as the dynamical decoupling technique, have been extended to open quantum systems. In this line, the results obtained by the authors contribute to the development of quantum control techniques by providing a method that allows optimization of the control parameter (e.g. the speed of moving the trapping potential).

Response:

We are happy that the reviewers have found our work relevant to current experimental endeavours in transferring atoms/ions through noisy environments using optical tweezers.

Comment:

However, the framework relies on the lowest order in the speed of the potential and the system-bath coupling strength, which seems to be basically a linear-response relation. Therefore, the validity of their methods, in particular to the case of fast non-adiabatic transport claimed in the abstract, is unclear.

Response:

As mentioned before, our theory provides a resummation of all second-order self-energy diagrams as described in details in [r1, r2]. We now explain in the revised SI Sec. V:

“The resulting dynamical equation for the amplitude $\alpha_o(t)$ [defined in (S19)], after this two-step Wigner-Weisskopf protocol, while retaining only the lowest order terms in the trap-speed and system-bath coupling (see Eq. (S28)) has the form:

$$\frac{d\alpha_o}{dt} = - \int_0^t ds \Sigma(t, s) \alpha_o(s), \quad (1)$$

where the kernel (self-energy) is given by, $\Sigma(t, s) = \left\{ \gamma_{n\epsilon}(t) \gamma_{n\epsilon}^*(s) + \Delta_{n\epsilon}^k(t) \Delta_{n\epsilon}^{k*}(s) \right\}$ (repeated indicies are summed). With the initial condition $\alpha_o(0) = 1$ we then solve the above equation exactly, to have

$$\begin{aligned} \alpha_o(t) &= \exp \left[- \int_0^t dt_1 \int_0^{t_1} dt_2 \Sigma(t_1, t_2) \right] \\ &= 1 - \int_0^t dt_1 \int_0^{t_1} dt_2 \left\{ \gamma_{n\epsilon}(t_1) \gamma_{n\epsilon}^*(t_2) + \Delta_{n\epsilon}^k(t_1) \Delta_{n\epsilon}^{k*}(t_2) \right\} \\ &\quad + \int_0^t dt_1 \int_0^{t_1} dt_2 \int_0^{t_2} dt_3 \int_0^{t_3} dt_4 \left\{ \gamma_{n\epsilon}(t_1) \gamma_{n\epsilon}^*(t_3) + \Delta_{n\epsilon}^k(t_1) \Delta_{n\epsilon}^{k*}(t_3) \right\} \\ &\quad \quad \quad \left\{ \gamma_{n\epsilon}(t_2) \gamma_{n\epsilon}^*(t_4) + \Delta_{n\epsilon}^k(t_2) \Delta_{n\epsilon}^{k*}(t_4) \right\} \\ &\quad + \dots, \end{aligned} \quad (2)$$

where in the last-step we have used the fact that $\Sigma(t_1, t_2)$ is symmetric in t_1 and t_2 .

Since $\Sigma(t_1, t_2)$ is quadratic in \dot{x}_o and H_{SB} , the above expression is a resummation of all second-order diagrams akin to a Dyson series [s5, s6]. This Wigner-Weisskopf non-perturbative approach is widely used in quantum optics [s5 – s9].

Hence, the survival probability $|\alpha_o(t_f)|^2$ in (S31) goes beyond linear response, since it takes into account the cumulative effect of an infinite sequence of second-order self-energies. This allows our method to be valid for fast non-adiabatic transport and strong system-bath couplings, as long as the relaxation is exponential.”

We also add in the manuscript:

“Then the expression for the survival probability at $t = t_f$, obtained from the Wigner-Weisskopf method, is a resummation of all second-order processes [12, 70], akin to a Dyson series (see SI V): ”

The above calculations and discussion also address the reviewers’ detailed comment (3), part of comment (4) and the first part of comment (5).

Comment:

There are several settings (see comments (1) and (2)) that seem unnatural in experimental situations and require physical justification or modification of the results.

Response:

We now explain in our detailed response to the reviewers' comments (1)-(4) and in the text we have added to Methods and SI, that the chosen settings are natural, realistic and justifiable in experimental situations.

Comment:

The presentation is quite technical, and the presentation could be further improved to increase the accessibility to general readers. These points may reduce the significance of the paper, and my general impression is that the paper is more suitable for specialized journals.

Response:

It is regrettable that the reviewers have found the previous presentation “quite technical”, and “more suitable for specialized journals” unlike the other reviewers who appreciated the generality and importance of this article. In fact, we were hoping for the reviewers' appreciation of our extensive efforts to present the complex, multi-faceted, uncharted domain that underlies this subject, simply and lucidly, with minimal technical detail (in the main text, as opposed to Methods and SI) so as to make it accessible to a broad readership. We are wondering: how can one underestimate the general nature of our original, unconventional and advantageous (compared to STA) approach to the timely and challenging problem of continuous-variable nonadiabatic, dissipative quantum transport?

We have now made an extra effort, in an attempt to follow the reviewers' suggestions, to further simplify the presentation and elucidate the points raised by the reviewers for the benefit of the readers. We hope that our extensive efforts will be appreciated by the reviewers and convince them of the significance and timeliness of our work and its suitability for Nature Communications.

Comment:

Detailed comments:

(1) As far as I understand, the fidelity (5a) is obtained by comparing the system-bath quantum state at time t : $|\psi_{SB}(t_f)\rangle = U_{SB}(t_f, 0)$ and the final target state $|\nu(t_f)\rangle = |n(t_f)\rangle \otimes |0_{\text{bath}}\rangle$, where $U_{SB}(t_f, 0)$ is the unitary operator by solving the Schrodinger equation. Because of this choice, only the case (i) in Fig. 1. C (i.e., purely non-adiabatic and no quanta exchange processes) contributes to the fidelity. However, the quantity of interest should be the fidelity of the reduced density matrix of the system $\rho_S(t_f) = \text{Tr}_B[\rho_{SB}(t_f)]$ to be in the target state $|n(t_f)\rangle$, defined by $\langle n(t_f)|\rho_S(t_f)|n(t_f)\rangle$, because this quantity expresses the probability of finding the system in the bound state of the trapping potential. The bath degrees of freedom is typically not accessible in experiments, and therefore the physical meaning of calculating the fidelity to find both the system in the target bound state $|n(t)\rangle$ and also the bath to be in the ground state $|0_{\text{bath}}\rangle$ is not clear. I think the results should be reformulated by using the fidelity by using the reduced density matrix of the system.

Response:

We thank the reviewers' for prompting us to elucidate this important point to which we now dedicate Methods A and explain in the main text as well as SI Sec. IV, VI. *Fidelity* is a measure of similarity of quantum states [see ref. [1] of the manuscript and refs. s23, s24 of

SI]. Different measures of quantum state fidelity are used in the literature depending on the context and the desired experimental outcome. There is no absolute measure of fidelity.

In particular, the fidelity measure we use, defined as the susceptibility of the dynamics to perturbations, is given by the overlap of the perturbed and unperturbed states at a particular instant – known as the Loschmidt echo: see [r1, r3]. In our problem, the perturbed state at $t > 0$ is $|\psi(t)\rangle$ which includes the effects of phonon-induced and non-adiabatic transitions (‘perturbations’), while the unperturbed state at $t > 0$ is $e^{-i\omega_0 t}|n(t)\rangle \otimes |0_{\text{bath}}\rangle := |\nu_1(t)\rangle$ (say), which includes no transitions induced by the ‘perturbations’ (i.e. “transition-less driving” of the wavepacket state). Thus, the Loschmidt-echo expresses the fidelity or the survival probability by $|\langle \nu_1(t) | \psi(t) \rangle|^2 = |\alpha_o(t)|^2$.

We beg to differ regarding the reviewers’ statement that our chosen fidelity measure is not accessible in experiments. In fact, Loschmidt echoes are measured experimentally, in quantum many-body dynamics and are used for probing dynamical phase transitions [r4–r8] (also see [r1] below along with ref. [74] of the revised manuscript). We now write in the manuscript:

“ It is important to note that Loschmidt echoes are routinely measured experimentally, in studying quantum many-body dynamics, and is often used as a probe for dynamical phase transitions [12, 74, 82–86].”

In our problem, wavepacket scattering by the phonon bath results in an entangled many-body state as we describe in the revised version:

“ At $t > 0$, the system-phonon scattering entangles *advanced*, *retarded* and *instantaneous* bound and excited states of the system wavepacket and the many-body bath states, which are required for momentum conservation (see SI IV).....”,

as explained in detail in the new Sec. IV in SI.

In quantum computing applications, the aim of our optimal transport strategy is quantum state preservation at the end of the transport. Thus, we intend to distinguish between the instantaneous (unscattered) and advanced/retarded (scattered) states at $t = t_f$. To this end we now write in Methods A:

“ Detection of a wavepacket state advanced by momentum k , implies corresponding collapse of the bath to an excited state having momentum $-k$. Similarly, detection of an instantaneous wavepacket state, ensures that the bath collapses to the ground state $|0_{\text{bath}}\rangle$ i.e. the survival probability in the instantaneous bound state (fidelity) is given by the Loschmidt echo which is here the probability of finding the combined system in the many-body state $|\nu(t)\rangle$ (see SI IV for details)....”

We modify Fig. 1 (C) in order to explain this and expand in Methods A:

“Only the diagrams in Fig. 1 C (i) corresponding to no quanta exchange (purely non-adiabatic) and virtual quanta exchange (purely bath-mediated) can result in the state $|\nu(t)\rangle$ at $t > 0$. The diagrams in Fig. 1 C (ii) and (iii) corresponding to real quanta exchange with the bath, result in advanced or retarded system states and corresponding excited bath states. A classical analog of this scenario is that of a catcher on a moving train that throws a ball vertically upwards. To an observer on the platform (in the laboratory frame), both the catcher and the ball undergo a horizontal displacement before the falling ball lands back in the catcher’s palm (in an instantaneous trap state). If instead the ball was launched

up in the train along with a finite horizontal momentum, the catcher may have to move backward or forward in order to catch the ball (in an advanced or retarded trap), thereby compensating for the additional momentum imparted. This reflects the fact that advanced/retarded states do not contribute to fidelity in the instantaneous state.

Finally, we address the reviewers’ suggested definition of fidelity, in the SI: Sec. VI.

“ The reduced density matrix $\rho_S(t) = \text{Tr}_{\text{bath}}(|\psi(t)\rangle\langle\psi(t)|)$ is a mixed state composed of advanced or retarded (gauge-transformed) and instantaneous system eigenstates, since $|\psi(t)\rangle$ is an entangled system-bath state [Sec. (IV)]. For mixed states (density matrices) [s23, s24], the most common fidelity measure is the Uhlmann-Josza fidelity $F(\rho_S(t), \sigma) = \left(\text{Tr}\sqrt{\sqrt{\rho_S(t)}\sigma\sqrt{\rho_S(t)}}\right)^2$, including its version when one of the density matrices is pure (Schumacher’s fidelity: $\langle\psi|\rho|\psi\rangle$), defined as the maximal transition probability between purifications of the two density matrices [s23 – s25]. Yet such a fidelity measure relies on the assumption that the two density matrices being compared, are derived from identically enlarged Hilbert spaces, which is not always the case [s23]. Moreover, being the maximum over purifications, such a fidelity measure over a reduced Hilbert space may be greater than the relevant fidelities for two pure states [s23]. In our problem, the instantaneous and scattered wavepackets $|n(t)\rangle$ and $e^{ikx}|n(t)\rangle$ may have a non-zero overlap in general, due to their finite width. However, the bath states $|0_{\text{bath}}\rangle$, $|1_{\text{bath}}^{-k}\rangle$ are all orthogonal, hence the many body states $e^{ikx}|n(t)\rangle \otimes |1_{\text{bath}}^{-k}\rangle$ and $|n(t)\rangle \otimes |0_{\text{bath}}\rangle$ are orthogonal. Therefore, tracing out the bath degrees of freedom reduces the distinguishability between scattered and unscattered contributions to the instantaneous states $|n(t)\rangle$. As a result, the corresponding Schumacher’s fidelity $\langle n(t)|\rho_S(t)|n(t)\rangle$ may include contributions from the scattered (advanced/retarded states) and exceed the true survival probability in the instantaneous state $|n(t)\rangle$, required for quantum state preservation.”

Comment:

(2) In line 217, I guess the authors mean that Eq. (7) is solved under the boundary condition $v(0) = 0$ and $v(t_f) = \text{some fixed value}$ (c.f. 0.5c, 1.5c). I think fixing the initial position and the trap-speed sounds reasonable in experimental situations. However, the choice of fixing the final trap-speed is unclear and needs physical justification. I would assume that fixing the final center position of the trap $x_o(t_f)$ is more natural and relevant to experimental situations.

Response:

The reviewers question the experimental relevance of the boundary conditions (BCs) we use, but these BCs are consistent with the basic premises of our method for finding the optimal trap-trajectory $x_o(t)$ and also with the BCs suggested by the reviewers. We have found the optimal trajectory by minimizing the exponent $J[x_o, \dot{x}_o]$ in Eq. 5(a), subject to an energy constraint $J_1[\dot{x}_o]$, in deriving the Euler-Lagrange Equation (see Eq. (11)). This is a standard optimization scheme where $x_o(t)$ acts as a generalized coordinate. As in the principle of least action, it amounts to finding $x_o(t)$ such that the functional $J[x_o, \dot{x}_o]$ attains a minimum (see [r9]) for the BCs $x_o(0) = \text{constant}$, and $x_o(t_f) = \text{constant}$, which are the same BCs suggested by the reviewers. Thus, there is no disagreement at all. On the other hand, for comparison with CDF, both initial and final trap-speeds must vanish (see ref. [72] of the revised manuscript). Otherwise, CDF becomes insufficient, as pointed out by the reviewers. Fixing $v(0)$, $v(t_f)$ and $x_o(0)$ then *uniquely* specifies the final trap position

$x_o(t_f)$ as we explain in the revised SI Sec. X:

“The Euler-Lagrange (EL) equation, Eq. 11, is a second-order integro-differential equation, which admits two arbitrary constants, specified by the two BCs: $x_o(0) = \text{constant}$ and $x_o(t_f) = \text{constant}$. In order to arrive at the simplified version Eq. (7) we have applied an additional time-derivative on both sides of Eq. (11). This comes with a penalty of requiring a third BC: Eq. (7) is second order in $v(t)$ and thus admits two independent BCs of $v(t)$. Together with the equation

$$\dot{x}_o(t) = v(t), \quad (3)$$

we then have three independent BCs in the solution for the optimal trajectory $x_o(t)$. Upon fixing the initial speed, $v(0)$, in addition to $x_o(0)$ and $x_o(t_f)$, we then **uniquely** specify the optimal trajectory including the final trap-speed $v(t_f) = \dot{x}_o(t_f)$.

On the other hand, for comparison with CDF, the useful BCs are to set the initial and final trap speeds to be 0 i.e. we choose $v(0) = v(t_f) = 0$ [s17]. Fixing the initial trap position $x_o(0)$ ($= 0$ in our case), we then **uniquely** determine the optimal trap-trajectory from Eq. (7) and (3), which inevitably fixes the final trap-position $x_o(t_f)$. One may instead choose to fix the initial and final trap positions along with the choice of $v(0) = 0$, which would then fix $v(t_f)$.

In order to show the superiority of our method over CDF, we have chosen the fixed final trap speed to be non-zero, a case where CDF fails but our method works well.”

We further write:

“For transport without a bath, $v(t)$ serves as the generalized coordinate for the EL optimization (see Eq. (8)). In this case, the BCs fix $v(0) = \text{constant}$ and $v(t_f) = \text{constant}$ instead of $x_o(0)$ and $x_o(t_f)$. Integrating over the optimal velocity profile we then get the optimal trajectory, which is **uniquely** determined by the initial position $x_o(0)$, which in turn fixes $x_o(t_f)$ as before. The corresponding survival probabilities are plotted in Fig. 3D. In this case, the control problem becomes solvable only after introducing the additional constraint on the total distance covered by the trap-center, $J_2[v]$, which fixes $x_o(t_f)$ while $x_o(0)$ is set by the BC.”

We have also added in the manuscript:

“This choice of boundary conditions uniquely determines the optimal trajectory $x_o(t)$, thereby fixing the final trap position $x_o(t_f)$ (see SI X, XII).”

To illustrate the equivalence of the two sets of BCs and hence the uniqueness of the optimal trajectory, we have analysed the case where $x_o(t_f) = \text{constant} \neq 0$, $x_o(0) = 0$ and $v(0) = \text{constant} = 0$ in Sec. XII of the SI. The equivalence of the optimal trajectories obtained from the two different sets of BCs is explicitly shown in Fig. S-4 of the SI, which we present below:

FIG. 1. (A) Comparison of the velocity profile $v(t)$ obtained from two boundary conditions. The profile $v(t)$ obtained for $v(0) = 0$ and $v(t_f) = 1.5c$ matches exactly with $v(t)$ obtained for $x_o(0) = 0, v(0) = 0$, and $x_o(t_f) = x_f = \int_0^{t_f} v(t)dt = 0.4612\xi$. (B) Same for $v(0) = 0$ and $v(t_f) = 0$ with $x_o(t_f) = x_f = 0.0582\xi$.

Comment:

(3) As the authors write in the supplementary material, the fidelities (survival probabilities) P are plotted using the analytical expressions Eqs. (28) and (29) [Eqs. (5a) and (5b) in the main text], which are valid in the leading order in \dot{x}_o and H_{SB} . In particular, if I understand correctly, Eq. (5a) should read $P(t_f) = 1 - J[x_o, \dot{x}_o] +$ higher-order terms, which is then approximated by $P(t_f) = \exp(-J[x_o, \dot{x}_o])$. If this is true, then by noting that, $\exp(-J)$ and $1 - J$ start to deviate from $O(J^2)$, for example, when P is below 0.9. Therefore, I suspect that some part of plots in Fig. 2 and Fig. 3 are outside the validity of the analytical expressions (5a) and (5b) (for example, AC-QUDIT methods: black solid curves with $0.4 < t_f/t_B$ in Fig. 3B, and Fig. 3C, STA methods: most of them in Fig. 3).

Response:

This comment does not correspond to what we do. We do not approximate “ $P(t_f) = 1 - J[x_o, \dot{x}_o] +$ higher-order terms by $P(t_f) = \exp(-J[x_o, \dot{x}_o])$ ”. As discussed in the response to a previous comment by the reviewers, $P(t_f) = \exp(-J[x_o, \dot{x}_o])$ follows directly from the Wigner-Weisskopf theory used in our work, which is a non-perturbative result obtained by resummation of all second-order diagrams. We emphasize that the plots in Figs. 2 and 3. are entirely within the regime of validity of the analytical expressions (5a) and (5b) of the manuscript.

Comment:

(4) The authors write in line 271-273 that “By contrast, $\delta_{n\epsilon}^k(t)$ remains invariant since CDF does not involve bath-induced transitions” but I think this is correct only if terms proportional to $g_k^2 \dot{v}^2$ are neglected.

Response:

We have not used the symbol $\delta_{n\epsilon}^k(t)$. Perhaps the reviewers mean $\Delta_{n\epsilon}^k(t)$, used in the text

(line 319 of the revised manuscript). The CDF term in the Hamiltonian is $-m\ddot{x}_\circ x$, as shown in line 295 of the revised manuscript, where x is the canonical position of the trapped particle in the laboratory-fixed frame. This term does not involve the bath operators. Hence, we have written “CDF does not involve bath-induced transitions”. On the other hand, $\Delta_{nc}^k(t)$ is defined by Eq. (S24) of the SI [using Eqs (S12, S13) of the SI] and also in line 204 of the revised manuscript as:

$$\Delta_{nc}^k(t) = d_{nc}^k(t)e^{-i\omega_{cnt}}e^{-i\Omega_k t} = \langle n(t)|e^{-ik\{x-x_\circ(t)\}}|\epsilon(t)\rangle e^{-ikx_\circ(t)}e^{-i\omega_{cnt}}e^{-i\Omega_k t}, \quad (4)$$

which is the transition matrix-element induced by H_{SB} . Since CDF does not involve bath operators, the system-bath coupling matrix-element remains invariant upon addition of a CDF. Our Wigner-Weisskopf non-perturbative technique includes terms proportional to $g_k^2 \dot{x}_\circ^2$. This does not change the fact that the phonon-mediated transition matrix-elements are not altered in the presence of a CDF.

The definition of CDF as an additional term added to the Hamiltonian, in order to achieve STA in isolated systems under the suitable BCs, makes it obvious that it is not designed to address bath-induced effects and thus, cannot alter the transition rates induced by an external bath.

Comment:

(5) *The authors write in line 406 that “Our general approach is rigorously proven to outperform STA protocols based on CDF for a broad class of transport trajectories and trapping potentials” but I think this sentence is misleading. I think the authors should write that up to leading orders in \dot{x}_\circ and H_{SB} , a condition to outperform CDF has been derived.*

Response:

We beg to differ. Our Wigner-Weisskopf non-perturbative technique is a resummation of all second-order self-energy diagrams. Thus, the condition we derive is not restricted to “leading orders in \dot{x}_\circ and H_{SB} ”.

Comment:

Related to the comment (2), the authors choose to fix the final trap-speed to be non-zero in all numerical calculations. This choice obviously makes CDF insufficient, even in the isolated case as plotted in Fig. 3D. Therefore, one might suspect that the comparison between AC-QUDIT and CDF discussed in this paper is limited to specific conditions in which CDF is insufficient. If the final center position of the trap is fixed, and the final trap velocity can be freely chosen, I guess that the plot would become very different, since one could choose $\dot{x}_\circ(t_f) = 0$ to make the CDF reproduce the adiabatic state at final times (note that for $0 < t < t_f$, the system does not necessary follow the adiabatic state).

Response:

See response to Comment (2) above, where the advantage of our method over CDF is detailed. We have made the final trap speed non-zero, a case where CDF fails but our method works! We now write in the manuscript:

“Importantly, the CDF method is designed to work only for $v(0) = v(t_f) = 0$ [27, 72]. We show, by contrast, that our optimal trajectory yields significantly higher survival probabilities than CDF, even for $v(t_f) \neq 0$, proving its superiority and wide range of applicability (see SI XI for $v(t_f) = 0$).”

The result for optimal transport with $v(t_f) = 0$, both in the presence and in the absence of an external bath, has been included in the revised SI which we present below:

FIG. 2. **Optimal transport with $v(t_f) = 0$** (A) Optimal trap speed for $m = 2.0m_B$, $\tilde{g} = 1.0t_B^{-1}$ and all other parameters as given in Methods C of the main text, with $\lambda = 1$ (black) and $\lambda = 0.07$ (green) (B) Corresponding survival probabilities. (C) Survival probabilities for non-adiabatic transport through vacuum ($\tilde{g} = 0$).

We explain in the SI:

“The BCs $v(0) = 0$ and $v(t_f) = 0$ yields an adiabatic optimal trajectory for $\lambda = 1$ with our chosen set of parameters, as shown in Fig. S-2 A (black curve). Even for such an adiabatic trajectory, we get survival probability nearly equal to that obtained with CDF, although our method yields slightly higher values (see Fig. S-2 B, black and red curves). To find a non-adiabatic optimal trajectory for this parameter regime, for fast transport of wavepackets, we can tune λ , which controls the total energy input for the transport. We find that for $\lambda = 0.07$ the optimal trajectory is non-adiabatic (Fig. S-2 A, green curve) with $v(t_f) = 0$ and the corresponding survival probabilities are much higher in our method than in CDF, both in the presence and in the absence of a bath (Fig. 2 B, C). This result confirms the optimality of our method over a wide range of parameter values.”

Moreover, we have provided a rigorous analytical expression showing the condition for the superiority of our method over CDF in Methods: D (see Eq. 16 therein), which does not depend on any particular choice of boundary conditions. This makes our claim much stronger than what the reviewers have interpreted.

The equivalence of the suggested BCs with our chosen BCs are discussed in detail in response to Detailed Comment (2) above.

Comment:

(6) Does the condition given in line 189 satisfied for the parameters used in Fig. 2, 3? Please add this discussion somewhere.

Response:

We thank the reviewers for this suggestion. We have added the following sentence in the revised manuscript:

“The corresponding results presented in Figs. 2 and 3 satisfy the condition $|v(t)| < v_s$ (see SI XII).”

We explain in the revised SI:

“The derived optimal velocity profiles all satisfy the condition $|\dot{x}_o(t)| = |v(t)| < v_s = |(\omega_{en} + \Omega_k)/k|$ necessary for the validity of the FDA and hence of the linearized Eq. (7) of the manuscript. To see this we compare the maximum value of the derived optimal velocity $|v(t)|_{\max}$ with the minimum value of v_s i.e. $v_s|_{\min}$. We note that v_s is a function of k and ϵ while $|v(t)|_{\max}$ is independent of k, ϵ . From Fig. S-2 we conclude that

FIG. 3. **Validity of the linearized Eq. 7:** (A) Plot of v_s and $|v(t)|_{\max}$ for $m = 2.0 m_B$ and $\tilde{g} = 1.0 t_B^{-1}$. The yellow surface shows v_s while the plane (black) indicates $|v(t)|_{\max}$. (B) Same for $m = 0.5 m_B$. All other parameters are as described in Methods C.

$$|v(t)|_{\max} < v_s|_{\min} \quad (5)$$

and hence

$$|\dot{x}_o(t)| = |v(t)| < v_s \quad \forall k, \epsilon, t. \quad (6)$$

Importantly, we have shown in Fig. S-1 that a numerical solution of the full non-linear integro-differential equation Eq. (11) with appropriate BCs using successive Picard iterations, converge to the solutions obtained from the linearized Eq. (7) derived using FDA. This result itself confirms the validity of the FDA and hence the condition $|v(t)| < v_s$ in our results. ”

Comment:

(7) From line 263 to 266: The authors write “This irreversible loss of fidelity cannot be avoided by merely making the velocity and acceleration vanish at the boundary points, as is usually done to achieve transition-less transport via STA [27,47].” Do the references [27,47] show that “This irreversible loss of fidelity cannot be avoided”, or they present the usual STA by “making the velocity and acceleration vanish at the boundary points”? If it is the latter case, please cite a paper or add additional discussions (numerical calculations) to clarify this statement.

Response:

Refs [27,47] ([27,72] of the revised version) of the manuscript present the usual STA by “making the velocity and acceleration vanish at the boundary points”. However, our problem deals with an unstable wave-packet that results in an irreversible leakage of a state into the continuum as pointed out in refs [9-11] of the manuscript in line 263 (refs. [8-10] in line 311

of the revised version). This can be understood by considering Eqs. (5a) and (5b) which describe this irreversible loss of fidelity as explained in the revised Methods F:

“The exponent on the r.h.s. of Eq. (5a), $J[x_o, \dot{x}_o]$ depends on the time-integral over the entire trajectory as shown in 5(b). The non-adiabatic contribution to $J[x_o, \dot{x}_o]$ (first term on the r.h.s. of Eq. (5b), transforms to Eq. (15) in the presence of a CDF (STA), which is essentially $\left| \int_0^{t_f} \Gamma_{n\epsilon}(s) ds \right|^2 \geq 0$. Likewise, the phonon-mediated term is $\left| \int_0^{t_f} \frac{L}{2\pi} \int dk \Delta_{n\epsilon}^k(s) ds \right|^2 \geq 0$. Since the coupling H_{SB} depends on the canonical position of the trapped particle x , making the trap velocity $v(t_f) = 0$ and trap-acceleration $\dot{v}(t_f) = 0$ at $t = t_f$ is not sufficient to make the phononic contribution to the irreversible loss of fidelity vanish. This shows that the irreversible loss of fidelity cannot be avoided. At best one might hope to make $\left| \int_0^{t_f} \Gamma_{n\epsilon}(s) ds \right| = 0$ by the CDF method. Yet Eq. (9) suggests that this would require $\gamma_{n\epsilon}(t) = im\dot{v}(t)D_{n\epsilon}e^{-i\omega_{n\epsilon}t}$ for all $t > 0$ and $\epsilon > 0$. This in turn requires that $v(t)\frac{\mu_{n\epsilon}}{\omega_{en}}e^{-i\omega_{en}t} = im\dot{v}(t)D_{n\epsilon}e^{-i\omega_{n\epsilon}t}$ i.e. $v(t) = v(0)e^{-i\int_0^t ds \frac{\mu_{n\epsilon}e^{-i\omega_{en}t}}{mD_{n\epsilon}e^{-i\omega_{n\epsilon}(t)}}$ for all $t > 0$ and $\epsilon > 0$. It is however impossible for a single velocity profile to satisfy an infinite set of equations of the form shown above for each value of $\epsilon > 0$.

This proves the fact that the irreversible loss of fidelity cannot be avoided by merely setting $v(t_f) = 0$ and $\dot{v}(t_f) = 0$ at the end point $t = t_f$.”

Comment:

(8) *Suggested references.*

System-bath control methods for non-Markovian systems:

<https://doi.org/10.1103/PhysRevX.10.031015>

Studies deriving exact analytical expressions and bounds for the fidelity for non-Markovian dynamics:

<https://doi.org/10.1103/PhysRevB.75.214308>

<https://doi.org/10.1103/PhysRevLett.97.200404>

<https://doi.org/10.1103/PhysRevLett.127.150401>

Response:

We thank the reviewers for suggesting these relevant references. We have now included the <https://doi.org/10.1103/PhysRevX.10.031015> in the revised manuscript. This reference discusses speed-ups to isothermality by tuning the system-bath coupling strength. It makes no attempt at minimizing non-adiabatic leakage rates that we explore in our work. In Appendix B of this reference, the resonant-level problem is solved under the more stringent assumption of a Markovian bath having a memory kernel that behaves like a delta function $\delta(t)$. In contrast, we have not made such assumptions.

The references <https://doi.org/10.1103/PhysRevB.75.214308>

<https://doi.org/10.1103/PhysRevLett.97.200404>

<https://doi.org/10.1103/PhysRevLett.127.150401>

discuss diagonal system-bath coupling for discrete variables that causes dephasing and do not account for system-bath entanglement. In such cases, Schumacher’s fidelity measure reduces to the overlap between pure-states since the reduced density matrix is pure. This cannot occur in the problem we describe.

Comment:

Typos:

Line 126: t should be replaced by 0

Line 161: $\langle r(t)|$ and $s(t)\rangle$ should be replaced by $\langle n(t)|$ and $|\epsilon(t)\rangle$.

Fig.2A caption: (green) and (blue) are opposite. They do not match the figure.

In addition, $v(t_f/t_B)$ should be replaced by $v(t_f)$. Similar typo for $P(t_f/t_B)$.

Response:

We thank the reviewers and have corrected these typos.

REFERENCES USED IN THE RESPONSE

- [r1] Boyanovsky, D., Jasnow, D., Wu, X-L. and Coalson, R. C., Dynamics of relaxation and dressing of a quenched Bose polaron. Phys. Rev. A, **100**, 043617, (2019).
- [r2] Boyanovsky, Daniel and Holman, Richard, JHEP, **2011**, 1–37, (2011).
- [r3] T. Gorin, T. Prosen, T. H. Seligman and M. Znidaric, Phys. Rep., **435** , 33 – 156, (2006).
- [r4] Xu, K. et. al., Probing dynamical phase transitions with a superconducting quantum simulator. Sci. Adv., **25** , eaba4935, (2020).
- [r5] Roberts, G. and Vrajitoarea, A. and Saxberg, B. and Panetta, M. G. and Simon, J. and Schuster, D. I., Manybody interferometry of quantum fluids Sci. Adv., **10** , eado1069, (2024).
- [r6] Cetina, M. et. al., Ultrafast many-body interferometry of impurities coupled to a Fermi sea. Science, **354** , 96–99, (2016).
- [r7] Braumüller, J. et. al., Probing quantum information propagation with out-of-time-ordered correlators. Nat. Phys., **18** , 172–178, (2022).
- [r8] Tonielli, F. and Chakraborty, N. and Grusdt, F. and Marino, J., Ramsey interferometry of non-Hermitian quantum impurities. Phys. Rev. Res., **2**, 032003, (2020).
- [r9] H. Goldstein, Classical Mechanics , Pearson Education India, (2011).

Response to Reviewers' Comments

REVIEWER #1

Comment:

I am happy with the corrections made following my suggestions. Also the replies to the other referee, in particular, reviewer No. 4, seem satisfactory. Some of the newly cited references have appeared in print now, so please cite the "originals", e.g. 37 and 60 Phys. Rev. A 111, L031301 (2025) and Phys. Rev. B 111, 064301 (2025). Moreover, some refs. have titles, other not, so please use a uniform presentation.

Response:

We are grateful that the reviewer has found our revision satisfactory, especially the changes made in response to reviewer # 4. Following the reviewer's suggestion we have updated references 37 and 60 in the revised manuscript. We have also corrected the non-uniform formatting of the references.

Comment:

With those minor changes the paper may be published.

Response:

We are happy that the reviewer has found our revised manuscript suitable for publication after minor changes.

REVIEWER #2

Comment:

The authors have addressed our observations in the revision, we thus recommend the publication of the article in Nature Communications.

Response:

We are happy that the reviewer has recommended publication of our revised manuscript in Nature Communications.

REVIEWER #3

Comment:

Response:

We thank the reviewer for co-reviewing the manuscript with reviewer # 4. We have addressed the points raised in our response to reviewer # 4, listed below.

REVIEWER #4

Comment:

I have read the revised manuscript and the response. The authors have adequately responded to most of my comments, and the clarity of the manuscript has been improved (see “Other minor points”).

Response:

We are happy that the reviewer has found most of our responses to be adequate. We are also glad that the reviewer has found the revised manuscript more accessible.

Comment:

Unfortunately, there is one point (the applicable regime of the proposed method) which I am not fully convinced. If the authors can clarify this point with adequate revision of the manuscript, the manuscript may be suitable for publication in Nature communications.

Response:

We concur with the reviewer on the desirability of a thorough discussion on the applicability of the proposed method which we have provided in the current revision. We believe that this addition fully addresses the reviewer’s queries and look forward to a positive recommendation of publication in Nature Communications.

Comment:

While I do understand that the expression for the fidelity is obtained by resummation of all second-order self-energy diagrams mentioned by the authors (meaning that the step from Eq. (1) to Eq. (2) in the response letter is exact), the starting point of Eq. (1) or (S28) requires “retaining only the lowest order terms in the trap-speed and system-bath coupling” as mentioned by the authors. Therefore, I am not fully convinced about the applicable regime of the expression (5a) and (5b) in the main text. For example, the authors write in the reply that “This allows our method to be valid for fast non-adiabatic transport and strong system-bath couplings, as long as the relaxation is exponential”, and “the condition we derive is not restricted to leading orders in trap-speed and system-bath coupling”, but the logic of these statements were unclear to me. Please explain why does the assumption of taking lowest order in the trap-speed and system-bath coupling to derive Eq. (1) is consistent with the above statement.

Response:

Indeed, the two quoted statements are mutually consistent. The reason is that our lowest-order approximation for $\frac{d}{dt}\alpha_o(t)$ in (S29) is very different from the standard lowest-order perturbative approximation for $\alpha_o(t)$: our approximation allows for the non-perturbative resummation of all second-order terms in $\alpha_o(t)$ as per the Wigner-Weisskopf theory (see Chapter XIII, Complement D_{XIII}, Sections 3 and 4 of [r1] and Chapter III of [r2]). The survival probability of the initial wavepacket given by Eqs. 5(a) and 5(b), has an exponential form obtained by this resummation which includes all powers of the second-order terms. Hence, it is “not restricted to leading orders in trap-speed and system-bath coupling”. In order to clarify this point we have added the following lines in the manuscript:

“In our non-perturbative approximation we retain only terms of leading order in \dot{x}_o and

H_{SB} , in the rate of change of the survival probability amplitude of the initial wavepacket [r1, r2]. This standard approximation in decay theory is valid whenever the coupling strengths (system-bath + non-adiabatic) are weaker than the inverse time-scales of the corresponding reservoir (bath or continuum) dynamics [r1–r8] (see SI V). Then the self-energy diagrams for the rate of change of the Loschmidt echo amplitude

We concur with the reviewer that the statement: “This allows our method to be valid for fast non-adiabatic transport and strong system-bath couplings, as long as the relaxation is exponential”, could be further clarified, which we have now done in SI V by adding the following explanation on the validity of our method, as requested by the reviewer.

“The rationale behind retaining only the lowest order contribution in the expressions for $\frac{d}{dt}\mathbf{A}(t)$ and hence $\frac{d}{dt}\alpha_o(t)$ is that the higher-order effects of systematic evolution become negligible upon averaging over a continuous and effectively infinite range of the bath or continuum frequencies [r1–r4, r6–r9]. This is true for couplings that are not strong enough to resolve the dynamics within the characteristic time-scales of this averaging. Since the transitions governed by \mathbf{U}_M in (S27) spread the wave packet throughout the continuum, we can estimate that effects beyond the leading order would be observable within a time-scale of the order of the inverse of the continuum width (energy uncertainty [r10]), which can be much smaller than the measurement resolution. For phonon-induced transitions, this characteristic time can be estimated as $\tau_B = (r_B/\xi)^2 t_B$ where r_B denotes the characteristic length-scale of inter-atomic interactions, t_B being the coherence time of impurity dynamics in BEC as before [r7]. Since r_B can be orders of magnitude smaller than the bath coherence length ξ , we have $\tau_B \ll t_B$ [r7]. Both non-adiabatic and phonon-mediated coupling strengths explored here are weaker than the inverse of the characteristic times discussed above so that our theory is accurate for times $t > \max\{\tau_B, \Delta\omega^{-1}\}$, $\Delta\omega$ being the continuum width.”

In order to further clarify the regime of validity of our results we have added the following paragraph in the Discussion:

“For transport through BEC, the time scales we have explored are shorter than the typical time for the emission of elementary excitations (polaron formation) in the BEC as well as their lifetimes, but longer than the time-scales on which the assumption of one-dimensionality of the BEC breaks down: the formation and lifetimes of elementary excitations exceed the coherence time of impurity dynamics in BEC, t_B [r7, r11], while the violation of one-dimensionality is manifest only at times shorter than the inverse transversal trapping frequency, which correspond to elementary excitations with energies high enough to break down the one-dimensionality [r12, r13].”

Comment:

As far as I can see from Eq. (5b), the quantities depend only on the system energy eigenenergies, eigenstates (and their derivatives), system-bath coupling operator, and the bath parameters, and therefore the information of the generically nonequilibrium time-evolution of the system-bath state is not captured. I guess this is why the authors can optimize the protocol by solving the EL equation, but on the other hand, this expression resembles me of linear-response relation where one could express quantities based only on adiabatic (e.g., energy eigenstates, etc) quantities and that optimization method for Hamiltonian parameters are possible.

Response:

Eq. 5(b) adequately describes the general non-equilibrium evolution of the joint system-bath state within its regime of validity discussed above. There is no loss of generality in the use of instantaneous eigenstates of the system, since they form a complete basis. So, it is legitimate to use this basis in Eq. 5(b), in the system-bath coupling or in any expression for the non-equilibrium evolution of the coupled system and bath.

Further, the definition of linear response [r14] is the dependence of the observables on the first power of the perturbation [r14] and is unrelated to the use of the instantaneous eigenstate basis.

The same is true of the optimization of any functional by solving the EL equation. This optimization is independent of the basis, or of the system-bath coupling Hamiltonian, and merely requires any smooth functional of some generalized coordinates. Whether such optimization would result in analytically tractable forms of the EL equation, depends on how complicated the functional is.

In our case, the maximized wavepacket survival probability (observable) Eq. 5(a) corresponds to the minimized exponent 5(b). We have directly minimized the exponent 5(b) to obtain the optimal survival probability (observable) 5(a).

Comment:

If there are some numerical plots obtained from numerical solution of the exact system-bath dynamics or using some non-Markovian numerical simulation methods for Gaussian baths, the validity of the methods proposed in this manuscript would be clearer (at least, the case of vanishing system- bath coupling strength should be doable).

Response:

We thank the reviewer for this interesting suggestion for future work, which would however be quite involved even in the case of vanishing system-bath coupling and is certainly beyond the scope of the present work.

To sum up, we have now clearly specified and thoroughly explained the regime of validity of our method as requested by the reviewer and are looking forward to a positive recommendation.

Comment:

Other minor points:

About the choice of the fidelity, I agree that if one could access all degrees of freedom, including the bath state, then the fidelity mentioned in the main text could be measured (as explained in the reply letter in terms of Loschmidt echo or similar methods). The authors have expanded physical discussions, including the scattered wavepacket discussion, which may clarify why the authors focus on the fidelity between final state and the instantaneous bound state with no bath excitations.

Response:

We are happy that the reviewer has found our discussion on the choice of fidelity measure, adequate.

Comment:

About the choice of the boundary conditions, I agree that fixing two values would determine

the whole quantities. I guess the following point clarifies the choice “In order to show the superiority of our method over CDF, we have chosen the fixed final trap speed to be non-zero, a case where CDF fails but our method works well”. I also wanted to understand how the plots Fig. 2 and 3 look like when the final position of the trap is fixed, but this may be optional.

Response:

We are happy that the reviewer has acknowledged that our choice boundary conditions is adequate. Since we have explicitly shown that fixing the final trap position would result in exactly the same optimal trajectory as obtained by fixing the final trap speed (see Fig. S-4), we opt not to replot quantities in Figs. 2 and 3, which are derived from this optimal trajectory and hence correspond to identical curves.

Comment:

The choice of vanishing final trap speed and comparison to CDF method is now much clearer, and I agree with the authors that their method is quite effective compared to the CDF method.

Response:

We are glad that the reviewer has found our revised presentation on the comparison with CDF much clearer and has acknowledged the effectiveness of our method compared to CDF.

REFERENCES USED IN THE RESPONSE

- [r1] Cohen-Tannoudji, C. and Diu, B. and Laloe, F. , Quantum Mechanics, Volume II, Second Edition, Wiley-VCH, Germany, (2020).
- [r2] Cohen-Tannoudji, C. and Dupont-Roc, J. and Grynberg, G., Atom-photon interactions: basic processes and applications, Wiley-VCH, Germany, (2004).
- [r3] Breuer, H-P. and Petruccione, F., The theory of open quantum systems, Oxford University Press, Oxford, (2010).
- [r4] Keitel, C. H. and Knight, P. L. and Narducci, L. M. and Scully, M. O., Resonance fluorescence in a tailored vacuum. *Opt. Commun.*, **118**, 143–153, (1995).
- [r5] Kofman, A. G. and Kurizki, G., Universal Dynamical Control of Quantum Mechanical Decay: Modulation of the Coupling to the Continuum, *Phys. Rev. Lett.*, **87**, 270405 (2001).
- [r6] Riera-Campenya, A. and Sanpera, A. and Strasberg, P., Quantum systems correlated with a finite bath: Nonequilibrium dynamics and thermodynamics. *PRX Quantum*, **2**, 010340, (2021).
- [r7] Nielsen, K. K., Ardila, L. A. P., Bruun, G. M. & Pohl, T. Critical slowdown of non-equilibrium polaron dynamics. *New J. Phys.* **21**, 043014 (2019).
- [r8] Gordon, G., Erez, N. & Kurizki, G. Universal dynamical decoherence control of noisy single- and multi-qubit systems. *J. Phys. B: At. Mol. Opt. Phys.* **40**, S75 (2007).
- [r9] Khal'fin, L. A., Contribution to the decay theory of a quasi-stationary state, *Sov. Phys. JETP*, **6**, 1053–1063 (1958).
- [r10] Kofman, A. G. and Kurizki, G., Acceleration of quantum decay processes by frequent observations. *Nature*, **405**, 546–550 (2000).

- [r11] Imambekov, A., Schmidt, T. L. & Glazman, Leonid I. One-dimensional quantum liquids: Beyond the Luttinger liquid paradigm. *Rev. Mod. Phys.* **84**, 1253–1306 (2012).
- [r12] Schmiedmayer, J. *Thermodynamics in the Quantum Regime: Fundamental Aspects and New Directions* One-dimensional atomic superfluids as a model system for quantum thermodynamics 823–851 (Springer, Cham, 2019).
- [r13] Torrontegui, E., Chen, X., Modugno, M., Ruschhaupt, A., Guéry-Odelin, D. & Muga, J. G. Fast transitionless expansion of cold atoms in optical Gaussian-beam traps. *Phys. Rev. A* **85**, 033605 (2012).
- [r14] Kubo, R., Statistical-mechanical theory of irreversible processes. I. General theory and simple applications to magnetic and conduction problems, *J. Phys. Soc. Jpn.*, **12**, 570–586 1957.